# Methods for identifying aged ship plumes and estimating contribution to aerosol exposure downwind of shipping lanes

Stina Ausmeel[1], Axel Eriksson[1,2], Erik Ahlberg[1], Adam Kristensson[1]

[1]Division of Nuclear Physics, Lund University, Lund, Box 118, 221 00, Sweden
[2]Ergonomics and Aerosol Technology, Lund University, Lund, Box 118, 221 00, Sweden

*Correspondence to*: Stina Ausmeel (stina.ausmeel@nuclear.lu.se)

**Abstract**

Ship traffic is a major source of aerosol particles, particularly near shipping lanes and harbours. In order to estimate the contribution to exposure downwind of a shipping lane, it is important to be able to measure the ship emission contribution at
various distances from the source. We report on measurements of atmospheric particles, 7-20 km downwind of a shipping lane in the Baltic Sea Sulphur Emission Control Area (SECA) at a coastal location in southern Sweden during a winter and a summer campaign. Each ship plume was linked to individual ship passages using a novel method based on wind field data and Automatic ship Identification System data (AIS), where varying wind speeds and directions were applied to calculate a plume trajectory. In a situation where AIS data is not matching measured plumes well or if AIS data is missing, we provide an
alternative method with particle number concentration data. The shipping lane contribution to the particle number concentration in Falsterbo was estimated by subtracting background concentrations from the ship plume concentrations, and more than 150 plumes were analysed. We have also extrapolated the contribution to seasonal averages and provide recommendations for future similar measurements. Averaged over a season, the contribution to particle number concentration was about 18 % during the winter and 10 % during the summer, including those periods with wind directions when the shipping
lane was not affecting the station. The corresponding contribution to equivalent black carbon was 1.4 %.

# 1 Introduction

Air pollution from anthropogenic activities, such as ship traffic, affects both human health and climate. Airborne particles cause negative health effects such as pulmonary and cardiovascular diseases, resulting in premature deaths and increased societal costs. Air pollution from combustion sources have an effect on climate due to emissions of greenhouse gases as well as particles with different optical properties and cloud forming capacities.

In order to reduce air pollution there are regulations on sectors with high emissions, for example the transportation sector. However, despite these regulations air pollution continues to be a serious problem. One sector that has gained relatively little attention in terms of emission control in the past is international shipping. The relative contribution from shipping to the total air pollution from transport is an increasing problem due to expected growth in shipping activity (Brandt et al., 2013; Corbett et al., 2007). One regulatory measure that has been taken to specifically reduce sulphur emissions from ships is the introduction of so called Sulphur Emission Control Areas (SECA), where the Baltic Sea SECA was one of the first to become established (Corbett and Fischbeck, 1997). In the International Convention for the Prevention of Marine Pollution from Ships (MARPOL) Annex VI, the main exhaust gas emissions of sulphur oxides (SOx) and nitrous oxides (NOx) are limited. Hence, the international Maritime Organisation (IMO) have regulated the fuel sulphur content in several steps, with a total decrease from 1.5% to 0.1% mass fraction between the years 2010 and 2015 in Sulphur Emission Control Areas (SECA). In 2016 it was decided that further reduction of the fuel sulphur limit is going to be implemented, with a cap of 0.50 % sulphur in fuel oil on board all ships from January 1st 2020. A recent report showed a compliance level to the sulphur regulations of 92-94 % during 2015 and 2016 in the region around Denmark (within the Baltic Sea SECA). (Mellqvist et al., 2017) Hence it is expected that most ships in the region are using fuels with a sulphur content of maximum 0.1 %. In addition to cleaner fuels, such as low-sulphur residual marine fuel oil, marine diesel oil (MDO), or liquefied natural gas (LNG), ships can comply by being equipped with scrubbers which remove the SO2 from the flue gas. The use of scrubbers was also observed in the region during our period of interest, by Mellqvist et al. (2017).

One way to characterize and quantify ship emissions is through measurements in coastal areas, downwind of a shipping lane. This makes it possible to register an increase in particle levels and the exposure to particles in this area when individual ship emission plumes pass the measurement station. With increasing distance from the emission source, the plume becomes more dilute and physically and chemically transformed due to atmospheric aging. In order to assess physicochemical properties but still capture features of the aged particles, which differ from the freshly emitted, it is therefore desirable to measure at an intermediate distance to the ships. Measurements of ambient aerosol particles are also important for an accurate assessment of the health effects, which depend on the actual exposure. This motivates measurements of the atmospherically aged ship aerosol particles from all types of ships affecting the coastal population. However, there are challenges associated with measuring aerosols from individual plumes further away from a moving point source such as a ship. Dilution will

eventually make it harder to distinguish from background levels, there can be an overlap of several plumes that intersect, and varying wind speed and wind direction makes it less obvious which ship is connected to which plume if the traffic is relatively intense.

All ships on international water with gross tonnage above 300 tons, cargo ships with gross tonnage above 500 tons, and all passenger ships are required to be equipped with a tracking system called Automatic ship Identification System (AIS). A ship sends out a position signal with individual International Maritime Organization ID and information about its type, size, country of origin, speed, etc. This data is collected every 6 minutes. AIS data in the Oresund region were used in this study to tie individual ship plumes to specific ships. AIS can be used as a tool in ship emission studies, commonly as a source for emission inventory used in models. This bottom-up method has been used and developed by many, e.g. (Jalkanen et al., 2009; Jalkanen et al., 2012; Liu et al., 2016; Beecken et al., 2015; Chen et al., 2018; Johansson et al., 2017; Marelle et al., 2016; Goldsworthy and Goldsworthy, 2015). AIS has also been used in connection to ambient plume measurements, to identify individual ship emission plumes. Alföldy et al. (2013) performed visual observations of ships at short distances in a port area and could connect these to live updates of ship positions. Ault et al. (2010) measured plumes and connected these to individual ships by using AIS ship positions and assuming transport with constant wind speed and wind direction. Lööv et al. (2014) used a similar method to locate plumes after emission, e.g. when doing airborne measurements within plumes further downwind of the ships. Diesch et al. (2013) also measured individual plumes and connected plume properties to ship properties, such as weight, using AIS, also at short distances (1-5 min downwind). Hence, AIS information has successfully been used in several applications, but for doing individual ship plume identification at longer distances where the plume might not travel along a straight path between emission and detection, other approaches might be needed that take into account the non-linear wind speed and direction. One example is the method of following ships either by aircraft or with a ship vessel up to a few kilometres behind the ship (Berg et al., 2012; Petzold et al., 2008; Chen et al., 2005; Williams et al., 2009; Lack et al., 2009). An advantage of this method is that the ships can be followed at different downwind distances, and can measure plume dilution and aerosol dynamics. However, it is an expensive method, and only a few ships can be followed due to budget and practical restrictions. Hence, this calls for a more feasible and cost-effective solution.

Particle number size distributions have been studied in atmospheric conditions previously, showing some variations in sizes and number of modes. This can be expected since many factors affect the emissions, such as engine operations, and the atmospheric transformation processes. For example, Jonsson et al. (2011) showed that size resolved particle number emission factors were largest around particle diameters of 35 nm, with smaller sizes observed for ships running on gas turbines than on diesel engines. Out of these particles, 36-46 % were non-volatile, and could contain some black carbon (BC). These measurements are from 2010, i.e. during the 1 % Sulphur limit within SECAs. Pirjola et al. (2014) showed that the number size distribution had two modes for fresh ship plumes, a dominating mode peaked at 20– 30 nm, and an accumulation mode at 80–100 nm. About 30 % of these were non-volatile, and it was also shown that the after treatment system affected the total particle number emission. These measurements are from 2010-2011. Diesch et al. (2013) observed a nucleation mode in the 10–20 nm diameter range and a combustion aerosol mode centred at about 35 nm. No particles with sizes above 1 µm were

observed. Six percent of the particle mass was due to BC. In the study by Diesch et al., AIS was used to link emission properties to ship properties, and they showed a decrease of most particle properties (incl. particle number concentration and black carbon) with increasing ship gross tonnage. Measurements on-board on a ship showed particle size distributions major peak at around 10 nm and a smaller peak at around 30−40 nm. Ca 40 % of the mass was non-volatile material, but particles below

10 nm consisted of only volatile material. (Hallquist et al., 2013) Westerlund et al. (2015) measured ship plumes from a stationary site and used AIS to characterise ships. Westerlund et al. found unimodal particle number size distributions for cargo and passenger ships, with the peak around 40 nm, while e.g. tug-boats emitted smaller particles. Since the measurements were carried out in a harbour area, as most of the other studies above, they could capture changes in emissions during e.g. acceleration of ships. These harbour measurements were carried out in 2010, i.e. also before the 2015 SECA implementation.

In another harbour area, Donateo et al. (2014) quantified the contribution of ship emissions to local total aerosol concentrations. The ship contribution to particle number was found to be 26 %. They could also see plume peaks in PM2.5, since measurements were done in a harbour area and plume peak concentrations were relatively high. A study performed in an Arctic region, showed a size distribution mode with peak around 27 nm during the first 6 hours of plume transport and later (>6 h) modes above 100 nm become more prominent. (Aliabadi et al., 2015) Here, the ship contribution to BC was estimated to be 4.3-9.8

15   %. Due to the clean Arctic environment and low background concentrations, the evolution of a ship plume contribution could be studied over time (0-72 h). In our measurements, we only observe ship plumes within the first hour of atmospheric transport. Dispersion modelling of ship plumes has shown that dilution and coagulation are important processes within the first hour after emission, reducing the number concentration by four orders of magnitude and one order of magnitude, respectively. (Tian et al., 2014) The decrease in particle number concentration is most rapid during the first minutes after emission. Our

measurements pick up ship emissions 15-70 minutes after emission (10-90[th] percentile). In general, for understanding the fundamental processes of ship emitted particles, detailed studies of individual engines or plumes from operating ships can be performed. However, for e.g. health effects the total contribution matters, so for investigating local particle contributions, a large set of ships must be studied.

We present a new revised method to identify individual aerosol ship plumes based on AIS data and non-linear wind

transport of the ship plume to a stationary coastal field site, which is several km downwind. The method has been tested on particle number concentration, particle number size distribution and black carbon mass. Also $CO_2$, $NO_x$, and aerosol mass spectrometry data is presented in the companion paper by Ausmeel et al. (*Ship plumes in the Baltic Sea Sulphur Emission Control Area: Chemical characterization and contribution to coastal aerosol concentrations*, manuscript in preparation, 2019b). The measurements were performed in Falsterbo, in southern Sweden, located downwind of a heavily trafficked

shipping lane in the Oresund Strait with a daily average of 73 and 63 AIS transmitting ships passing in winter and summer respectively, and which connects the Atlantic and the Baltic Sea. The distance from the shipping lane to the site corresponds to an average transport time of between 15 and 70 minutes (10-90[th] percentile) for the ship plumes. The measurements took place during the winter (Jan-Feb) and the summer (May-Jul) of 2016. With the new revised plume identification method, we can detect several tens of plumes in a day with favourable wind conditions. We also show how particle number concentration

data can be used when AIS data is failing or missing, to identify individual ship plumes, however without information about which ship it is.

We identified and calculated the contribution as well as the particle size distribution of individual ships by subtraction of background concentrations from the identified plume particle number concentrations. In addition, we have developed and described a new method to calculate the contribution of aerosol properties when the plume cannot be visually distinguished from background concentrations due to noisy data and relatively weak contribution at this fairly long distance from the shipping lane. This method has been tested on equivalent black carbon (eBC) concentrations. eBC is black carbon mass concentration derived from optical absorption measurements and a mass absorption cross-section (MAC) value. (Petzold et al., 2013) In our measurements, the MAC value for the 880 nm wavelength was 7.77 $m^2g^{-1}$. (Drinovec et al., 2015) The duration of a eBC plume is based either on the available ship plume identification from the AIS and wind data, and plume evolution of particle number concentration data, or only on particle number concentration data when AIS data is not available. For the aerosol properties for which ship plume concentrations could be calculated, a daily and seasonal average contribution for the entire fleet could be estimated.

## 2 Instrumentation set-up and experimental site

The location of the sampling site was on the Falsterbo peninsula in south-western Sweden (55.3843 N, 12.8164 E) (Fig. 1). The measurement location is within a SECA covering the Baltic Sea. The main shipping lanes, which pass to the west and the south of Falsterbo, are about 7-20 km away from the measurement site. The surrounding area is mainly made up of open coastal landscape, with roughly 250 m of reed and sand dunes separating the measurement site from the open water of Oresund. There are few buildings and activities nearby. To the north, east and south of the site there is a golf court and to the east of the site, i.e. not between the shipping lane and the site, there is a workshop connected to the golf court. South of the site there is a lighthouse, housing a weather station run by the Swedish Meteorological and Hydrological Institute (SMHI). Vehicles and machinery passing the measurement site were considered when analysing the data.

A PM10 aerosol sampling inlet was mounted at a height of about 4 m above ground, on top of a mobile trailer housing the instruments. The trailer was air-conditioned and kept at an indoor temperature of about 20° C. Fig. 2 shows a sketch of the complete measurement setup and the flow configuration used in the Falsterbo measurement campaigns. The total particle number concentration was measured with a condensation particle counter (CPC, TSI 3775 or TSI 3025) with a sample time of 30 s. In addition, a custom built scanning mobility particle sizer (SMPS) (Wiedensohler et al., 2012) was used to measure the particle number size distribution in the electrical mobility diameter range 10.5-532 nm (DMA, Hauke type medium, custom built; CPC 3010, TSI Inc., USA) (Svenningsson et al., 2008). The time resolution was two minutes per scan. Particle size distribution in the micrometre range (0.54-19.8 μm) was measured with an aerodynamic particle sizer (APS 3321, TSI Inc. USA). Equivalent black carbon (eBC) content was measured with optical absorption methods, using a seven wavelength Aethalometer (model AE33, Magee Scientific) (Drinovec et al., 2015) with a sample time of one minute. Data from several of

the instruments in Fig. 2 will be presented in a companion article, Ausmeel et al. 2019b. The chemical composition of sampled particles was evaluated with a Soot Particle Aerosol Mass Spectrometer (SP-AMS, Aerodyne Research Inc.). (Onasch et al., 2012) In addition to the AMS measurements, black carbon (BC) content was measured with optical absorption methods, using a seven wavelength Aethalometer (model AE33, Magee Scientific) (Drinovec et al., 2015) and a 637 nm Multi Angle Absorption Photometer (MAAP, Thermo Fisher Scientific) (Müller et al., 2011), both with a sample time of one minute. A potential aerosol mass oxidation flow reactor (PAM OFR) (Kang et al., 2007; Lambe et al., 2011) was alternately connected before the AMS, SMPS, and Aethalometer to simulate atmospheric aging. For the gaseous aerosol compounds, CO2 concentration was measured with a non-dispersive infrared gas analyser (LI-COR LI840) and SO2 was measured using a UV fluorescent monitor (Environnement S.A AF22M). $CO_2$ concentration enhancements due to ship plumes were below the detection limit of the monitor used, which means that emission factors likely cannot be calculated for ship plumes 7-20 km downwind of the shipping lane. In summary, for the MAAP (detection limit, DL, of $< 50$ ng m$^{-3}$), APS (DL 0.001 cm$^{-3}$), $CO_2$ (DL $< 1$ ppm), and $SO_2$ (DL $< 1$ ppb) monitors, the concentrations from ship emissions were at all times undistinguishable from the background levels. These data sets were not analysed further.

During the summer campaign, the aerosol flow for certain instruments (Fig. 2) was dried using either diffusion or membrane (Nafion) driers. The particle losses in the membrane dryers due to diffusion were determined by laboratory measurements. For 100 nm particles the losses were in the range 0-10%, and for 10 nm particles the losses were about 5-20%. Specifications about the dryers and losses can be found in Table 1 and Fig. 3. These losses are used to correct the size resolved scanning mobility particle sizer (SMPS) data. Table 1 presents the specifications for each dryer used in the summer campaign to dry the aerosol particles before sampling with some of the particle instruments. Letters A-C correspond to the dryers shown in the illustration of the Falsterbo measurement setup in Fig. 2. The flows for which the losses are characterised were the same flows as used in the field measurements. The aerosol used for the characterization was polydisperse ammonium sulphate in lab room air. The resulting losses, as a fraction of the total particle concentration, are shown as function of particle size in Fig. 3. In addition, corrections for particle losses in the sampling line was calculated using the Particle Loss Calculator tool (Von der Weiden et al., 2009), and were applied to the SMPS size distributions but not for the other instruments.

## 3 Methods for identifying ship plumes and estimating ship contribution

### 3.1 Ship plume identification and analysis

To confirm the contribution of ship plumes to particle and gas concentrations in Falsterbo, the time when each ship plume should influence the Falsterbo site was estimated with revised method based on Automatic ship Identification System (AIS) position data as well as wind direction and wind speed data from Falsterbo lighthouse Swedish Meteorological and Hydrological Institute weather station. (SMHI, 2017)

Only ships passing by in the area limited by a rectangle with geographical coordinates [(55.16 N 12.45 E) (55.56 N 12.45 E) (55.56 N 13.00 E) (55.16 N 13.00 E)] were included in the analysis (Fig. 1). The data for the ship positions were

available with a time resolution of 6 minutes, and the wind data were available with a one-hour time resolution. Since a higher time resolution was needed to identify ship influence at the measurement station, the ship positions, wind directions and wind speeds were linearly interpolated to a one-minute time resolution.

For each interpolated one-minute ship position, wind trajectories were calculated describing how the wind travelled from the ship at time 0 ($t_{emission}$) towards the Falsterbo station, until the wind approached the Falsterbo station at time instance $x$ minutes ($t_{arrival}$). The minimum distance between the Falsterbo station and the wind trajectory defined $t_{arrival}$. Each ship passage in the rectangle contained several of these minimum distances since we used all one-minute ship positions when calculating $t_{arrival}$. The shortest distance among this subset of each ship passage was chosen as the final minimum distance. This method is similar to the method by Lööv et al. (Loov et al., 2014) and Ault et al. (Ault et al., 2010). However, in those studies, the distances between the ships and the station were much shorter. Hence the authors could use a wind direction and wind speed that did not change with time along the trajectory between the ship and the station, while in this study, the wind direction and wind speed is varying between $t_{emission}$ and $t_{arrival}$, which is a novel method of estimating ship plume positions over greater distances.

When the wind was not arriving from the sea, the ships did not influence the measurements. Ship passages were defined to influence the Falsterbo station only if the minimum distance between the wind path at $t_{arrival}$ and the Falsterbo station was smaller than 500 m. The effect of ship emissions on the particle concentrations at Falsterbo were strongest and clearest for the number concentration and particle number size distribution data. Hence, each $t_{arrival}$ when a ship should influence Falsterbo measurements, was compared to the actual measured data. In theory, it is possible that the wind direction is changing as the ships sail past the measurement station, meaning that we can potentially miss the maximum concentration in ship plumes, and only record the lower concentrations at the tails of the ship plumes. However, in almost all cases in our data set, the wind is stable enough during each ship plume passage at the station. This means, we fetch entire ship plumes, from the lowest concentrations in the plumes to the maximum concentrations in the plume.

There is a significant uncertainty in finding the $t_{emission}$ and $t_{arrival}$, since the wind data was interpolated to one-minute values from a one-hour resolution, and due to the fact that the wind trajectory path was calculated based on the wind data from Falsterbo. In reality, the wind speed and wind direction along the ship plume travelling from the ship towards Falsterbo could occasionally be significantly different, especially for ships, which are sailing far away from the Falsterbo station. Despite this uncertainty, each $t_{arrival}$ matched very well with increases in particle number concentrations during winter. A majority of $t_{arrival}$ are within 5 minutes of the actual concentration peaks, as illustrated in Fig. 5. However, when two or more ships influence the Falsterbo station almost at the same time, it is hard to distinguish which individual ship is contributing most to the increase in particles. During summer, the method to match AIS data with ship plume peaks yielded a lower agreement presumably due to less stable meteorological conditions during summer, e.g. more turbulence, and sea breeze. Nevertheless, the method worked surprisingly well even for this period for the few number of plumes identified. In the end, however, the AIS method was not used during summer, since AIS data were not available more than a few days due to errors in the AIS database.

Even for periods when AIS data was not matching plume times well, or when AIS data was missing from the AIS data base, particle number concentrations could be used to identify ship plumes instead. This required that there were no other interfering particle number concentration sources, or that these could be distinguished from the ship plumes. The number concentration data was then used to identify the plume time period, since particle number concentrations were always above detection limit for all ship plumes, and the time resolution was large enough to clearly identify the shape of the plume peak. However, all increases in particle number concentration were not a result of ship emissions, but rather land going vehicles passing the measurement site. These could be recognized and excluded. Normally, the land going vehicles were influencing the particle concentrations for a minute or shorter, while the ship plumes that influenced the particle concentrations could last for several minutes up to about 20 minutes. Note that the alternative method of identifying plumes with number concentration is not giving information about which ship passed by the measurement site due to lack of AIS data, unless there are other ways of collecting this information.

## 3.2 Calculating the contribution of ships to aerosol number concentration and other properties

For an identified ship plume peak, the contribution from this plume was estimated by calculating the area under the peak after subtraction of background concentrations. An example of a measured ship plume and illustrations of these calculations are shown in Fig. 4. In Fig. 4, the particle number concentration is clearly elevated during a few minutes during a period of relatively constant background concentrations. The estimated time of arrival of the plume, based on wind and AIS data (as described previously), is marked with a star and confirms the measurement of a ship plume and could provide further information about the ship, if desired. Due to the frequent appearance of ship plumes in Falsterbo, the background concentration was calculated as the average concentration of two intervals, one just before and one just after the ship plume, as seen in Fig. 4.

One alternative way to calculate plume contribution by subtracting plume from background is the method used by Kivekäs et al. (2014). The authors extracted particle background concentrations by taking the 25th percentile values of a sliding window of a few hours for the particle number concentration time series. This is an appropriate automatic method to use on large data sets of ship plumes. The ship lane in the Kivekäs study was between 15 and 60 km away from the station. During periods with sharp increases or decreases of background concentrations, this method did not yield acceptable results, and these periods had to be manually controlled for errors and removed from the final data analysis. However, the Kivekäs method was not possible to use in Falsterbo due to the frequent plume events and the relatively high number concentrations in the plumes, which affected the background values for the sliding window method.

If a measured concentration of some aerosol parameters is noisy or the plumes are similar in concentration to the background, it is still possible to use AIS or particle number concentration to identify plumes and calculate their contribution. This could be the case when particle mass concentrations in the ship plumes are generally low. E.g., a plume peak is not clearly distinguished, as depicted in Fig. 4 for eBC mass concentrations. However, based on the identification from the AIS and the estimation of the plume duration from particle number concentration data, the effect of the plume on the other aerosol

parameters could be investigated. The contribution from a ship to such an aerosol parameter was calculated in the same way as described above, by subtracting the adjacent background concentrations from the concentration during the plume period. The start and end time of the plume was assumed to be the same as measured by the particle counter.

Beside the contribution to aerosol concentrations in each plume, there is also a possibility to estimate the contribution from ships at a coastal location during an extended period of time, like a day, a season, or a year. This can be accomplished by multiplying the average plume contribution with the number of ships that have passed during the current period and including a factor of how large fraction of the time the wind was passing over the shipping lane towards land. We estimated the daily and seasonal contribution of ships ($f_i$) to the particle concentrations at Falsterbo, in addition to background levels, using the equation

$$f_i = \frac{c_{ship}}{c_{bgr}} \cdot \frac{n_{ship,i} \cdot t_{plume,av}}{t_i} \cdot w_i \tag{1}$$

where $c_{ship}$ is the average ship plume concentration, $c_{bgr}$ is the average background concentration for the chosen time period ($i$), $n_{ship,i}$ is the number of ships passing during this period (based on AIS data, independent of wind direction), $t_{plume,av}$ is the average ship plume exposure duration, $t_i$ is the length of the time period $i$, and $w_i$ is the fraction of the time during which the wind is blowing over the shipping lane to the location of interest (defined by a reasonable wind sector for the location).

## 4 Results

### 4.1 Plume identification

To demonstrate how the ship identification with the AIS method worked, Fig. 5 shows an example of a time series from the CPC for a few hours of sampling during wintertime. Fig. 5 also displays the times when the ship plumes were expected to arrive at the measurement station based on AIS and wind data, as described in section 3. The particles from the ship plumes are seen as relatively short and intense peaks, generally matching well with the expected plume passages. The average plume duration was ten minutes. All ships identified with the AIS system resulted in an increase in size-dependent particle number concentration when these measurements were available. The method to infer when the ship plume should affect measured concentrations at Falsterbo agreed excellently during winter considering that the wind speed and direction measurements had a one-hour resolution and that these parameters were only measured at Falsterbo and not along the air mass trajectory. In summer, this agreement was reasonable, but less certain than in winter, which might be due to more turbulent winds, and local meteorological factors such as sea breeze. This shows that the method has a potential to work for many different shipping lanes. All plumes passing the measurement site are observed in the particle counters, that is the fraction of observed plumes predicted by AIS-trajectories is in principle 1. We miss some plumes in the individual ship analysis, due to too frequent and overlapping plume passages. We estimate the analysed fraction to ca 0.4 for the ship traffic near Falsterbo. The analysed fraction depends on the plume duration as well as the frequency of ships. With an average plume duration of about 10 minutes, it also means that the plume peak maxima should be separated by at least 10 minutes to be able to correctly calculate plume

contributions. For studies which do not require information about individual ships, but rather about total ship contribution, the number of missed ships is very low and can be due to temporary AIS malfunction or military vessels passing (they do not transmit AIS). The highest uncertainty of the timing of the plume is introduced through the wind trajectories between emission and measurement site. Regarding the uncertainty of the attribution of a ship-ID to a plume, this is depending mainly on the frequency of ship plumes at the specific location in combination with the wind trajectories. If the plumes from two ships arrive about the same time to Falsterbo station, we cannot be absolutely sure which ships contributed to which plume concentrations. In that case, we only know that two ships did contribute to elevated concentrations. Also, if these plumes are superimposed on top of each other, we are still not able to calculate the individual ship contribution. We choose only to calculate plume contribution for plumes which peaks are about at least 10 minutes apart in order to avoid plume superposition, since average plume duration is about 10 minutes as stated in the manuscript. In this case, the ship-ID identification is always appointed to the correct plume. We have seen that the timing accuracy of the ship-ID with the actual plume contribution is better (lower) than 7 minutes (95 % CI). Since, we choose only plumes or Ship-id data which are at least 10 minutes apart, this uncertainty has no effect on attributing a ship-ID to the correct plume.

As an example of what AIS information can be used for, the properties of the ships identified in Falsterbo during the winter campaign are shown in Fig. 6. The distribution of ship weight, length, breadth, and average speed, as well as the distance from the emission source to the measurement site (in distance, km, and in transport time, minutes) are shown. The units of the parameters have been adjusted so that all values fit within a similar range in the plot. The linear distance from the ship to the measurement site at the time when the ship contributed to the pollution at the site is denoted "ship to site / km" and given in km, and the transport time of the wind between the ship emissions and the site is denoted "ship to site / min", and given in min. Note that the wind does not necessarily travel along a straight line between the ship and the station if the wind direction is changing, which is considered in the calculation of the "ship to site / min".

No relation was found between emission and ship properties or transport, therefore the data presented is not normalized for e.g. weight or transport time but presented as it was measured at the measurement site. A variety of vessels are passing Oresund Strait and Falsterbo. The most common ones are cargo ships, tankers and ro-ro ships (roll-on/roll-off) and others are trawlers, dredgers, reefers and fishing vessels. The production years of the ships ranged from 1965-2015, with a majority from the 1990's and 2000's.

### 4.2. Results of plume contribution calculations

The contribution of ship traffic to the air pollution at a coastal location was estimated for more than 150 ship plumes. Measurements were carried out with a similar setup during winter (January–March) and summer (May-July) of 2016. All instrument variables were not available for the entire measurement periods and the wind direction was not always favourable for measuring ship plumes. In total, there were about three weeks with optimal data from the winter campaign and two weeks from the summer campaign.

For the calculation of how ships contributed to the particle number concentration, plumes were restricted to the following conditions: 1) identified by AIS, 2) clearly distinguishable from the background in the CPC time series, and 3) not overlapping with other plumes. This resulted in 109 (CPC) and 113 (SMPS) plumes from the winter campaign and 61 (CPC) and 8 (SMPS) plumes from the summer campaign used for further calculations. The number of plumes identified by the SMPS in the summer is much lower than identified by the CPC due to a non-functioning SMPS system in periods. Also, periods during which the SMPS was sampling aerosol through a potential aerosol mass oxidation flow reactor (PAM-OFR) were also excluded in this analysis. Finally, there were in general fewer plumes identified in the summer than in winter due to lack of AIS data in summer and since the winds at Falsterbo less frequently arrived from the shipping lane during the summer measurement campaign. A summary of the meteorological conditions during the measurements can be found in Table 2.

According to the methods described in 3.2, we calculated the individual contributions from the observed ship plumes, both for particle number concentration and for eBC mass concentration, as well as the estimation of a daily and seasonal contribution at the specific location according to Eq. 1. This calculation was based on AIS data, which showed an average of 73 and 63 ships passing per day in winter and summer respectively. Together with the average plume duration (10 min), this indicates that the Falsterbo site is affected by ship emissions 51% of the time in the winter, and 44% in the summer, when the wind blows from the Oresund Strait. Based on historical wind data from the last 20 years (Swedish Meteorological and Hydrological Institute), the wind intercepts the shipping lanes in Oresund Strait about 70% of the time in both summer and winter, which was used together with particle concentration measurements to estimate the seasonal contribution from ships. Example of plume contributions, both individual, daily, and seasonal, are shown in Table 3. For each of the $n$ number of measured ship plumes, a contribution is calculated. The table shows the median of these values, as well as the 25th and 75th percentile. A general observation was relatively large differences between ships, hence a larger number of observed plumes is preferred for a better estimation of the local ship emission contributions.

Regarding the uncertainty in the plume particle number contribution, the relative statistical error of the CPC count is related to the total count $N$ by $\sqrt{(N)}/N$. Hence, the particle counter has a very high precision. During our sample length of 1 s, the number $N$ was typically above 1000 cm$^{-3}$, and for an entire plume the total count was much higher. The uncertainty of the total concentration given by the instrument does also depend on the uncertainty in the sample flow rate, since the concentration output is equal to "$N$ / (flow rate · sample time)". We assume a flow rate uncertainty of maximum 10 %. So for example, with a concentration of 1000 particles cm$^{-3}$ and a flow of 1 litre per minute, the uncertainty becomes 10.5 % (when adding in quadrature). For an entire plume, the statistical error is even smaller and hence the total uncertainty in particle number concentration is basically equal to the uncertainty in the sample flow. There is also a bias in concentrations due to losses of the smallest particles in the sampling line. That is, we measure lower concentrations than the ambient since this effect removes particles. Diffusion losses have been corrected for in the size distributions. But since we did not have SMPS and CPC data with same time interval (2 min vs 1 s) we cannot know exactly the losses for the CPC.

Despite the fact that the plumes were not clearly visible in the eBC time series, due to the low contribution to mass, a significant increase in BC was observed during identified plume events. The seasonal contribution of ship emitted eBC is on

average only 1.4 ± 0.6 % of the total measured eBC at Falsterbo. Due to the noise of the eBC data as depicted in Fig. 4, individual eBC plume contributions are occasionally negative. However, a t-test was performed on this data, which showed that the value of the eBC plume contribution was significantly higher than zero with a p-value of 0.000030. Artificial eBC-data without noise were also created, and random noise was applied on this data, which were of the same amplitude as real noise of eBC data to test whether noise in data creates a systematic over- or underestimation of the plume contribution data. The test showed that the noisy eBC data is not creating an over- or underestimation of plume contribution, and hence this plume contribution should be robust. The same analysis was done on $CO_2$ concentrations as for eBC, where plumes were also not visually distinguishable from background levels. Hence, at the distance from the shipping lane in this field study in Falsterbo, the plume $CO_2$ concentrations were too diluted upon arrival at the measurement site to be distinguished from ambient levels. Therefore, it was not possible to calculate emission factors of particles for the ship plumes. For regional and global models, it would be useful with emission factors for the slightly aged ship plumes as well as for fresh ones, which is obtained in e.g. laboratory engine studies and harbour measurements. If emission factors are to be determined for slightly aged ship plumes it is possible that a shorter distance than our 7-20 km is preferred and a sensitive $CO_2$ monitor (limit of detection below 0.1 ppm) is needed.

The mean and median particle number size distribution for the ship emission plumes in Falsterbo are shown in Fig. 7. The distributions were calculated by averaging the number concentration in each SMPS size bin for 113 ship plumes for the winter campaign and 8 ship plumes from the summer campaign. A log-normal function (Hussein et al., 2005) with several modes was fitted to the average and median size distribution plumes for the winter and summer seasons. For the log-normal function, only particles with an electrical mobility diameter larger than 15 nm and smaller than 150 nm are considered due to uncertainties and losses for other sizes. The log-normal parameters are listed in Table 4. Four or five modes are used in the log-normal fit of the average size distribution plume since it seems that the typical size distribution contains a smaller and a larger sized nucleation mode (mode no. 1 and 2, < 30 nm diameter), and a smaller and larger sized Aitken mode (30 to 100 nm diameter). A majority of the ships do not produce the lower sized nucleation mode, why the median size distribution does not contain this first mode. The other modes are often all present at the same time and the larger particles could arise due to coagulation in an aerosol with a high concentration of smaller particles or due to emissions of relatively large primary soot particles. The uncertainties for the size distribution are large for the particles in the upper Aitken mode (80 to 100 nm diameter) and the accumulation mode (>100 nm diameter) due to low numbers counted in the SMPS and also due to large variation between individual ships. The Pirjola et al. (2014) study shows that the particle number size distribution has two distinct modes for fresh ambient ship plumes, one in the nucleation mode (<30 nm diameter), and one in the Aitken mode (30-100 nm diameter). If the number size distribution is remade into a volume size distribution, also an accumulation mode becomes visible (>100 nm diameter). The current study also contains these modes. In addition, due to the individual variability between ship plumes in the current study, even two Aitken modes are discernible in the log-normal fitted size distributions. A few of the ships have a distinct accumulation mode, and for this reason, the average size distribution also contains this log-normal fitted mode. The data is significantly corrected for particle losses in sampling tubing especially for the nucleation mode sizes (<30

nm diameter), which makes a second log-normal nucleation mode below 15 nm diameter appear in the log-normal fitted size distributions. Lab engine measurements also show such a mode in the Anderson et al. (2015) study, when higher sulphur content fuel was used, which stimulated new particle formation. Hence, in total, there are 3 to 5 log-normal modes fitted to the median and average particle number size distributions (Table 4).

The number size distributions in Fig. 7 show that essentially all particles in the average and median ship plume have an electrical mobility diameter below 100 nm, most of them around 20-40 nm. Similar results have been shown in laboratory and on-board measurements (Kasper et al., 2007; Betha et al., 2017; Isakson et al., 2001; Kivekäs et al., 2014). There have also been observations of larger particle diameters, in the µm-range, e.g. (Fridell et al., 2008). In our study, the APS instrument did not show any contribution to µm-sized particles from ships at the current distance from the shipping lane. The APS has a

high sensitivity for single particles but did not measure that ship plumes contained significant particle number concentrations above background concentrations for particles larger than 0.5 µm diameter. Since we did not observe any particles larger than a few 100 nm in Falsterbo, this could be a suggestion that the larger particle modes are absent or negligible after the recent SECA sulphur regulations. Particles larger than the upper detection limit of the APS (ca 20 µm) were not measured and could have been present, but in that case likely deposited on the way to land.

The size distribution of the average plume shows higher concentrations than that of the median plume, both for the summer and the winter data. This is due to the high contribution of some ships skewing the results. Due to higher and noisier background particle concentrations in the summer (Table 3), and the lack of AIS data, it is possible that plumes with relatively low particle number concentrations were not distinguishable from the background, and hence the selection of plumes in the summer might have been biased towards the more polluting ship plumes.  Also, the difference in sample size should be noted

here, 113 good plumes observed during winter and 8 during summer. This difference depends mainly on instrument malfunction, unfavourable wind directions, and lack of AIS data. From the available data, there is however an indication that the number and the size of the particles from ships are somewhat larger in summer. This seasonal difference could possibly be explained by secondary PM formed by atmospheric aging, which is expected to be more significant in the summer, but more measurements are needed to confirm this.

Due to the distance to the shipping lane, the ship emissions were diluted enough to have concentrations below the detection limit of some instruments. In order to capture the relatively short plume events, the time resolution could not be too long either, making the detection limit of some instruments higher.

## 5. Recommendations and concluding remarks

The AIS method to identify which ship influenced exposure on land and to identify individual ship plumes from measurements about 10 km downwind of ship lanes proved to be very exact for the winter data and worked relatively well for summer data. We know that the method to observe individual plumes on top of background concentrations does not work for all ships at the distance 25-60 km downwind of a shipping lane. (Kivekäs et al., 2014) There, only a fraction of the plumes were distinguishable. In contrary, at our Falsterbo site, there were no such issues with the plume identification method. Hence, the method can be expected to work at least up to 10 km and getting worse towards 60 km. This is true for particle number concentration measurements (with a CPC) but not for mass concentration measurements. So, to be able to detect plumes at maximum distances a particle counter is of importance. Considering the wind data was available only with a one-hour resolution, the plume identification worked well. Availability of wind data with better time resolution does not seem to be necessary at this specific site. Although at longer distances between the ship lane and the station, this can potentially be an issue and it would be advantageous to have meteorological data with better time resolution. When AIS data was missing for one reason or the other, the particle number concentration detected with a condensation particle counter also proved to work very well to identify ships, although could not give the information about which ship it was.

The method to estimate plume contribution from individual ships proved to be straightforward for the clearly visible ship plumes at the measurement station. For the eBC concentration, the plume identification was less straightforward since the plume signal was very low relative to the noise level. For many plumes, no increase in eBC was observed with the bare eye. We still used the already identified plumes to calculate the contribution to eBC. A very low, but still significant plume contribution could be calculated. Even if the proposed method yields non-significant plume contributions for a specific parameter, this does not mean that the method does not work. Rather it means that ship emissions do not contribute to significant exposure inland for this parameter, and that the detection capabilities of the instrument do not allow for detecting this non-significant contribution. The calculation was done using the precise time of the plume incidence observed in the particle counter. This was also a surprisingly robust method without systematic biases due to the noise. Dispersed background levels of BC were about 0.2 $\mu g\,m^{-3}$. The ship emission particles, which were clearly seen by number in the plumes, do probably contain soot since they are from a combustion source but the mass becomes difficult to detect due to the small particle sizes. It could therefore be valuable for future measurements of ship emitted BC to use a measurement method which does not require much BC mass for detection, such as single particle incandescence.

Since the particle counter always yielded visible and smooth plumes at the downwind station, it is recommended always to bring a particle counter when doing these kind of measurements, even if it is not of main interest to estimate particle number contributions. Namely, it might turn out that AIS data is erroneous or missing, and the particle counter is needed to define plume time and background to calculate the plume contributions for instruments with high noise and low time resolution. Since the ship plumes at 10 km downwind or longer from the ship lanes have only a few minutes up to about 20 minutes' duration, it is also recommended that the time resolution of the instruments one brings is not worse than one minute. For a

scanning particle sizer, like the SMPS, one should consider the scan time in comparison to the plume duration, and possibly add a mixing volume to not get rapid changes in the aerosol particle concentration during a scan.

These and other measurements have shown that the number of particles < 30 nm diameter is substantial, even for relatively aged ship plumes. The estimation and correction for particle losses are therefore crucial to be able to assess the true size-dependent particle concentrations, especially when the sampling line to instruments is relatively long. It is our recommendation to place the particle counter (CPC) close to inlet, and further to use as short sampling line as possible with minimum diffusion losses when performing these kind of studies in general.

The current method of stationary measurements of downwind plumes from a shipping lane has turned out to be very cost-effective compared to aircraft or ship vessel chasing experiments, and can fetch a much higher number of ship plumes. Hence, we also urge to use it for economic and pragmatic considerations when studying relatively aged ship plumes for a high number of ships. In future studies of detailed individual ship plumes and the emission sources, it should be considered whether the particle emissions depend on ship engine power used. It is possible to estimate the engine power required by a ship, using the total power of the ships, their design speed, and actual speed through the propeller law. (Moreno-Gutiérrez et al., 2015) This can then be compared to particle number concentration emissions, but also particle mass emissions and gaseous emissions. With the method presented in this paper, it is possible to collect information on a very large sample of ships for these kinds of investigations.

Before performing the measurements with the new method, it is important to investigate the meteorological situations at the current measurement site. For example, during sea breezes, local wind measurements could indicate that shipping lane emissions should reach the measurement station, whereas in reality they might not. Care should be taken to account for these periods when the meteorological data will give erroneous results. However, these meteorological phenomena do not take place all the time, hence these specific meteorological conditions will not disqualify any chosen measurement site with the current proposed method. Again, these uncertain wind conditions make it very important to bring a particle counter to register shipping plumes. If the particle counter does not register any ship plumes during a selected time period, this indicates that winds from the ships are not reaching the measurement station, despite that the local wind measurements are suggesting otherwise."

Beyond providing ambient aerosol data from a SECA from summer and winter measurements, the data from this study can also be used to validate process models simulating aging processes of particle number size distributions as well as long distance transport along meteorological air mass trajectories in Lagrangian process models. In addition to particle number concentration and eBC, the method was applied in the companion paper, Ausmeel et al. 2019b, focusing on other aerosol properties. and regional or global scale air quality and climate models could use this kind of data to validate modelled ship contribution in certain grid cells.

**Author contribution**

A. Kristensson designed the experiments and all authors carried them out, with A. Kristensson being the project leader during the winter campaign and S. Ausmeel during the summer campaign. A. Kristensson developed the model code. S. Ausmeel analysed the plume and aerosol data and prepared the manuscript with contributions from all co-authors.

**Competing interests**

The authors declare that they have no conflict of interest.

**Acknowledgments**

This study was financed by the Swedish research council FORMAS (project no. 2014-951). The Crafoord Foundation (projects no. 20140955 and 20161026), is acknowledged for the contribution to the MAAP instrument funding and the funding of the postdoc position for the current studies. Fredrik Windmark at the Swedish Meteorological and Hydrological Institute (SMHI) is acknowledged for helping to provide AIS ship positioning data. Mårten Spanne, Paul Hansson, Henric Nilsson, and Susanna Gustafsson from the Environment Department at the City of Malmö are acknowledged for helping preparing and setting up the measurements at Falsterbo. Dr. Kirsten Kling of DTU and Dr. Antti Joonas Koivisto of NRCWE are acknowledged for helping with the summer campaign, and Fredrik Mattsson and Anna Hansson, for helping with the winter campaign. Thank you also to Håkan Lindberg and the personnel from Falsterbo golf court and Lennart Karlsson from Falsterbo bird watching station who were willing to prepare a place for our measurement trailer, and to the County Administrative Board of Skåne and Vellinge municipality for giving permission to measure in Flommen Nature Reserve.

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

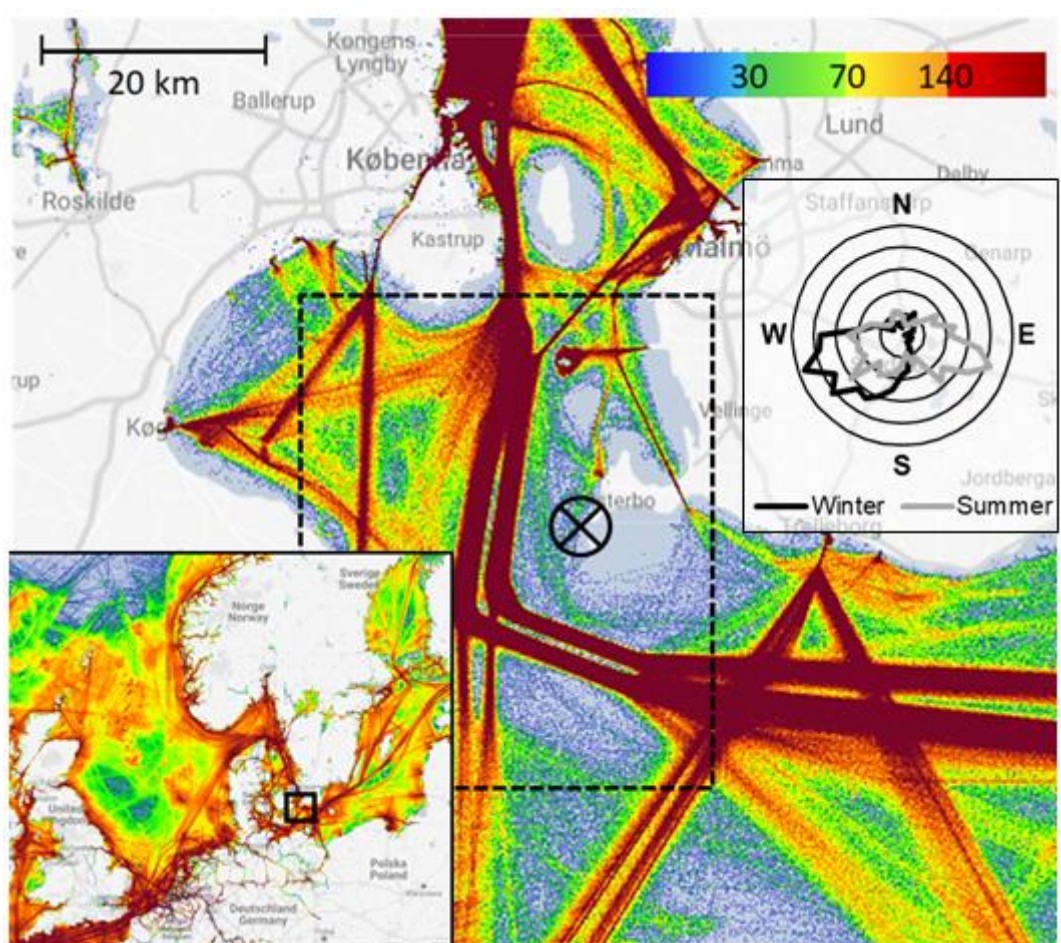

**Figure 1. Location of the measurement station (circle with cross) at the Falsterbo peninsula together with ship traffic density, the colour bar indicates an approximate number of distinct vessels passing per day per km² (www.marinetraffic.com, 2016). Dashed square shows the area in which AIS positions are considered for ship identification. Inserted to the right is the wind direction pattern during the winter (black) and summer (grey) campaign respectively.**

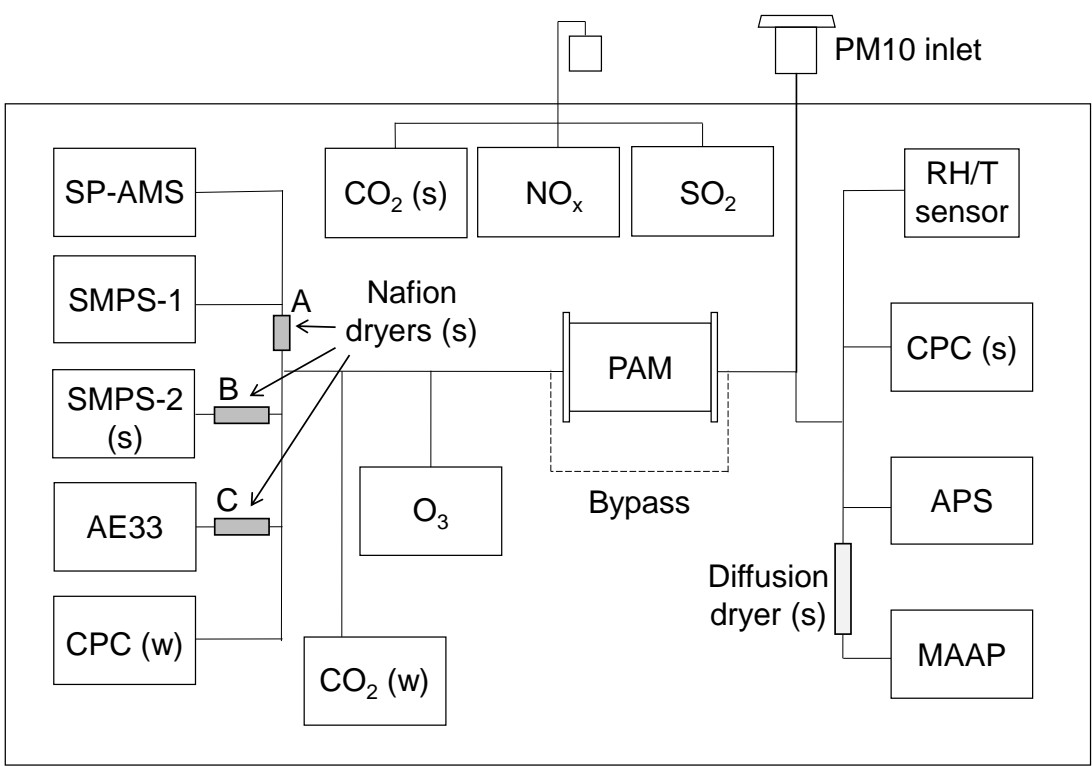

**Figure 2. Measurement setup. The symbol (s) indicates configuration used only during the summer and (w) only during the winter. Dashed line shows the bypass flow excluding the PAM oxidation flow reactor from the sampling line. For the membrane (Nafion) dryers, the letters correspond to the size dependent particle penetration for each drier shown in Fig. 3.**

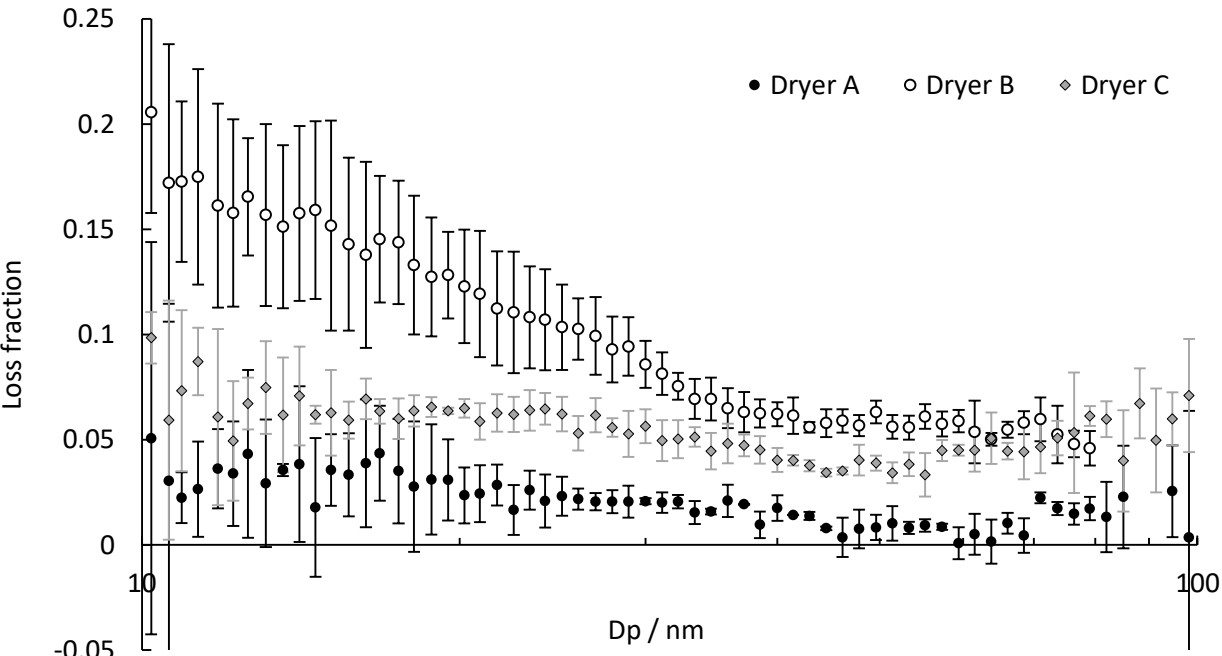

**Figure 3. Fraction of total particle concentration lost due to diffusion in three dryers, as a function of particle diameter, $D_p$. Error bars indicate one standard deviation from 2-3 measurements.**

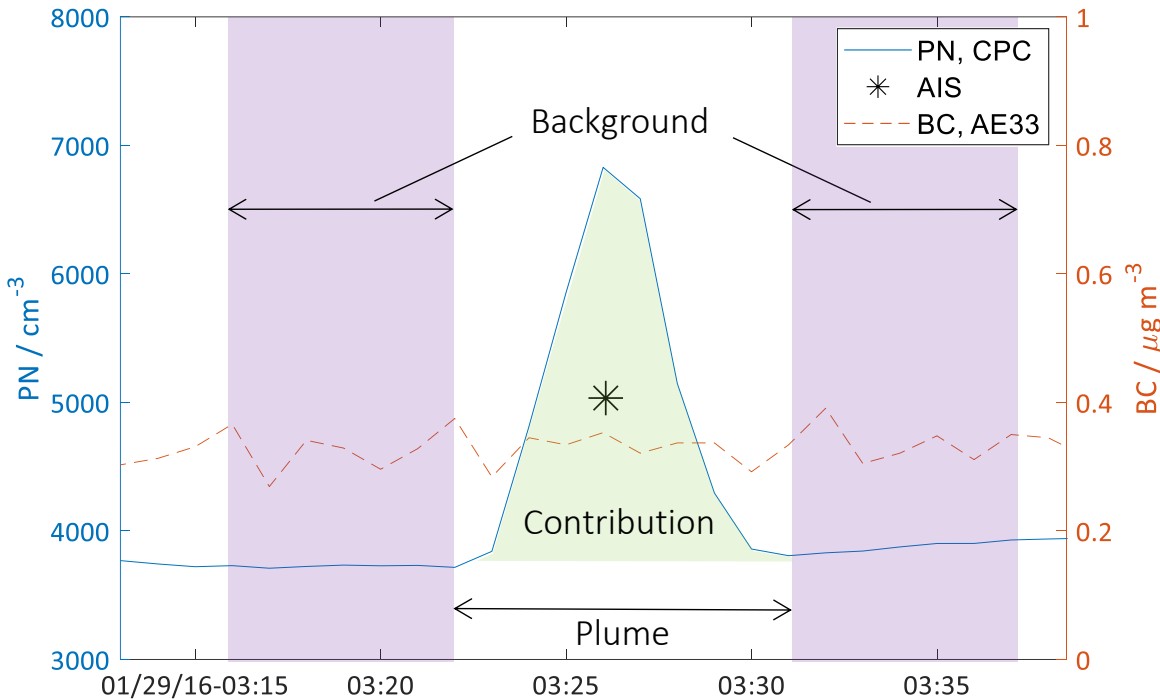

**Figure 4. Illustration of method of calculating aerosol contribution of individual ship plumes. Particle number concentration measured by a CPC (solid blue) and black carbon measured by an Aethalometer (AE33, dashed orange) during ca 25 minutes of ambient sampling, and calculated time of arrival of the aerosol plume based on AIS and wind data (star). Plume duration is estimated by observation and background concentrations are based on six plus six minutes of adjacent data. The average of the background is subtracted from the plume concentrations to obtain only ship emission contribution.**

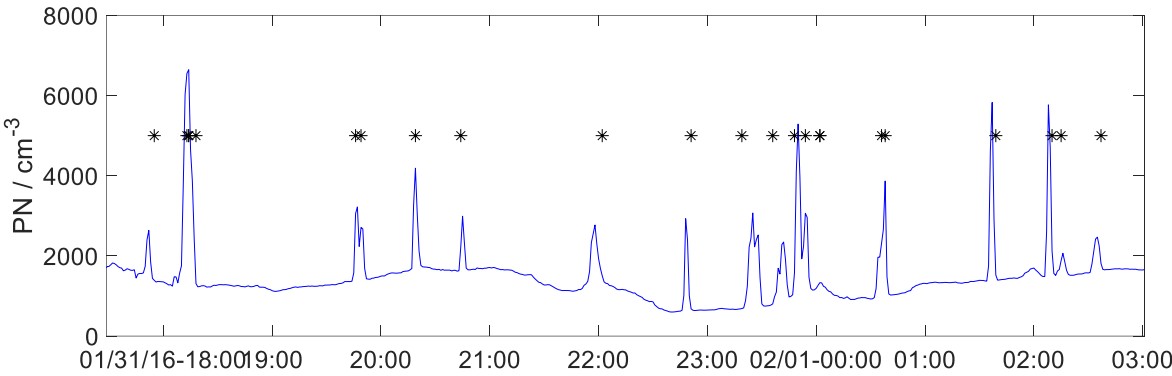

**Figure 5. Particle number concentration measured with a CPC, and calculated incidents of ship plume passages (stars) determined with AIS and meteorological data, versus time (31 Jan. – 1 Feb. 2016), from measurements at the coastline in southern Sweden during an episode with westerly winds blowing from the Oresund Strait to the coastal station Falsterbo. The concentrations are those of the total aerosol, i.e. background concentrations are not subtracted.**

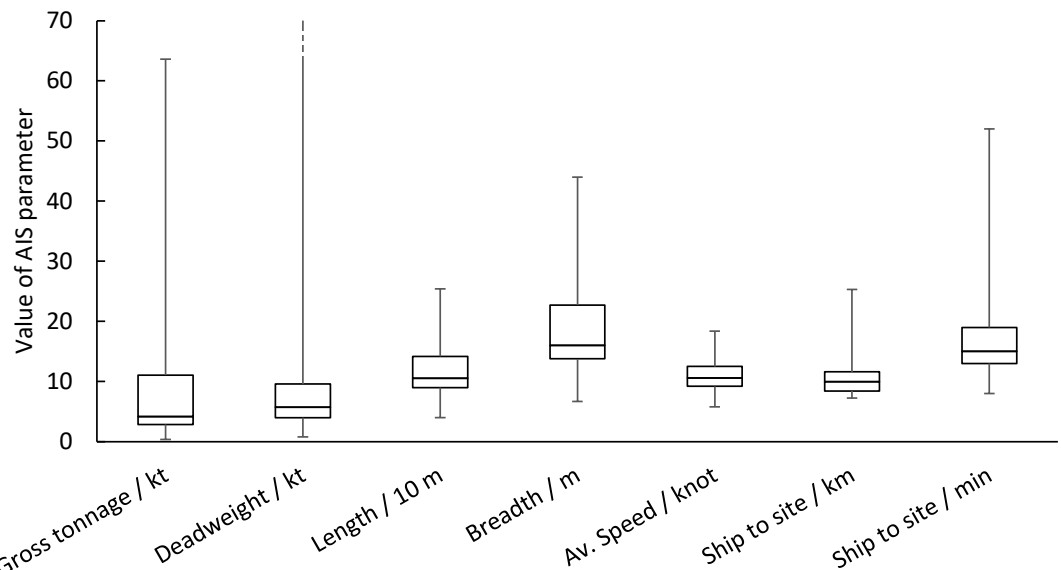

**Figure 6. AIS ship information and calculated plume travel data for the 113 plumes evaluated from the winter campaign in Falsterbo. The boxes show median, 25th and 75th percentile, and whiskers show minimum and maximum value. The maximum deadweight of 140 kt is out of range.**

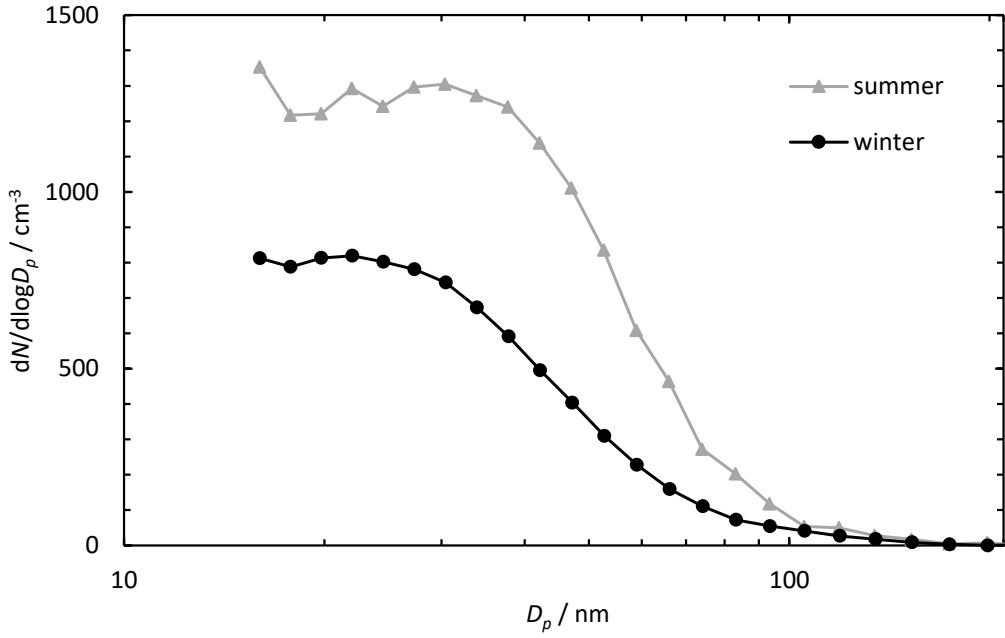

**Figure 7. The ship contribution to the average size distribution of particles (diameter, $D_p$, from 15 to 200 nm), measured with an SMPS during winter (n = 113) and summer (n = 8) respectively. Ambient background concentrations have been subtracted for each plume event and correction for particle losses in the sampling has been accounted for.**

**Table 1. Specifications of dryers used in Falsterbo, letters A-C correspond to the driers in Fig. 2.**

| Dryer | Serial no. | Dryer length (Wiedensohler et al.) | Flow used (lpm) |
|---|---|---|---|
| A | MD-110-12E-S (082913-02-18) | 30.5 | 1.1 |
| B | MD-110-24S-4 (1060301) | 61 | 0.3 |
| C | MD-110-24S-4 (0860108) | 61 | 3 |

**Table 2. Meteorological conditions during measurement campaigns, average, lowest and highest values.**

| Parameter | Winter (18/1 - 3/3) | | | Summer (16/5 – 7/7) | | |
|---|---|---|---|---|---|---|
| | Min | Av ± stdv | Max | Min | Av ± stdv | Max |
| Temperature (deg. C) | -5.2 | 2.5 ± 2.4 | 6.6 | 9.2 | 17.0 ± 2.9 | 25.9 |
| RH (%) | 63 | 87.8 ± 7.9 | 99.0 | 38 | 77.7 ± 11.8 | 99.0 |
| Wind speed (m/s) | 0.3 | 7.5 ± 3.4 | 17.0 | 1.0 | 6.4 ± 2.8 | 15.0 |
| Sunlight (h/day) [a] | 0 | 1.93 ± 2.44 | 9.37 | 0 | 7.96 ± 4.57 | 15.6 |
| Precipitation (mm/day) | 0 | 0.8 ± 1.4 | 5.9 | 0 | 1.6 ± 3.8 | 31.8 |

[a] Direct sunlight, i.e. not cloudy.

**Table 3. Contribution of particle number concentration and eBC mass concentration to local air quality, from two measurement campaigns at the Falsterbo coastal site.**

| | Variable (instrument, $D_p$-range) | Background concentration [a] | Ship plume concentration [a] | | | Average contribution from shipping lane | | $n$ [a] |
|---|---|---|---|---|---|---|---|---|
| | | | 25th perc. | Median | 75th perc. | Daily (%) [b] | Seasonal (%) [b] | |
| Winter | N / cm$^{-3}$ (CPC [c], 4 nm-10 μm) | 1320 | 440 | 750 | 1130 | 25±5 | 18±4 | 109 |
| Winter | N / cm$^{-3}$ (SMPS [d], 15-532 nm) | 1200 | 340 | 700 | 1080 | 26±9 | 18±7 | 113 |
| Winter | eBC (ng m$^{-3}$) [e] | 210 | 0 | 9.9 | 20 | 2.0 ± 0.8 | 1.4 ± 0.6 | 100 |
| Summer | N / cm$^{-3}$ (CPC, 4 nm-10 μm) | 2610 | 600 | 860 | 1180 | 14±3 | 10±2 | 61 |
| Summer | N / cm$^{-3}$ (SMPS, 15-532 nm) | 2530 | 710 | 1470 | 1930 | 26±10 | 18±7 | 8 |

[a] The background particle concentrations (Background conc.) and the particle contribution due to ships (Ship conc.) to number concentration (N) are shown as absolute values. Each value represents a median (or percentile) of a number of plumes (n) and are calculated from the ship plume peaks average concentration (i.e. concentration per unit time).

[b] "Daily" values refer to days with wind directions where ships affect Falsterbo (mainly westerly) and "Seasonal" values refer to the average contribution observed at each campaign extrapolated over one season, including all wind directions.

[c] Condensation Particle Counter

[d] Scanning Mobility Particle Sizer

[e] Based on Aethalometer data (880 nm).

**Table 4. Lognormal fit parameters for the average and median size distribution of the detected ship emission particles, during winter (n=61) and summer (n = 8) respectively.**

| Parameter | Winter | | Summer | |
|---|---|---|---|---|
| | Median size distr. | Average size distr. | Median size distr. | Average size distr. |
| $N_1 \pm \Delta N$ (cm$^{-3}$) | - | $603.66 \pm 615.34$ | $1302.72 \pm 1054.28$ | $916.82 \pm 731.82$ |
| $N_2 \pm \Delta N$ (cm$^{-3}$) | $584.75 \pm 290.75$ | $890.06 \pm 351.06$ | $942.05 \pm 278.05$ | $775.03 \pm 329.97$ |
| $N_3 \pm \Delta N$ (cm$^{-3}$) | $222.33 \pm 167.67$ | $214.66 \pm 171.66$ | $293.57 \pm 84.57$ | $603.19 \pm 188.81$ |
| $N_4 \pm \Delta N$ (cm$^{-3}$) | $7.79 \pm 11.21$ | $35.46 \pm 32.60$ | $17.49 \pm 23.51$ | $13.58 \pm 25.42$ |
| $N_5 \pm \Delta N$ (cm$^{-3}$) | - | - | $9.68 \pm 7.62$ | $22.74 \pm 21.56$ |
| GMD$_1$ (nm) | - | 9.95 | 10.74 | 14.13 |
| GMD$_2$ (nm) | 19.04 | 21.46 | 29.84 | 27.49 |
| GMD$_3$ (nm) | 34.17 | 41.52 | 51.82 | 46.65 |
| GMD$_4$ (nm) | 86.29 | 98.04 | 84.16 | 84.31 |
| GMD$_5$ (nm) | - | - | 125.73 | 112.16 |
| $\sigma_1$ | - | 1.30 | 1.35 | 1.35 |
| $\sigma_2$ | 1.51 | 1.60 | 1.35 | 1.35 |
| $\sigma_3$ | 1.39 | 1.45 | 1.22 | 1.35 |
| $\sigma_4$ | 1.37 | 1.32 | 1.10 | 1.10 |
| $\sigma_5$ | - | - | 1.13 | 1.27 |

