# Peer review of "Methods for identifying aged ship plumes and estimating contribution to aerosol exposure downwind of shipping lanes"

_Atmospheric Measurement Techniques, 2018_

## Referee Comment (RC1) · Anonymous Referee #1 · 4 Mar 2019

The study correlates measured particle data with individual ships in the region of the measurement station, using AIS data to identify ship and ship position, and measured wind patterns to calculate expected plume arrival time allowing assigning of plume peaks to individual vessels. Hourly meterological data were interpolated to 1 minute. Other components of ship plumes did not yield significant peaks. The reference list is rather short and the Introduction too brief. It should cover expected characteristics of particles and other components of ship plumes (including particle size distribution and plume dispersion) as well as SECA regulations including fuel S content and possible use of scrubbers. P 8 last para – why is there no correlation with ship size? While the interpolation of hourly wind data to one minute yielded quite good correlation

between ships identified by AIS and arrival of detected plumes at the measurement station, more detailed meteorological data might have allowed estimation of the extent of dispersion of the ship plumes and the location of the core of the plumes relative to the measurement station. This may have allowed better correlation between ship size and detected plume. For instance, the main part of a plume from a large ship might pass some distance from the measurement station with only the more dispersed outer extent of the plume registering at the station. This is likely the reason for the poor correlation between ship size and size of particle number peaks. At Line 30 it is stated that there was no information on engine operation and fuel. However, this can be derived from the AIS data according to well established methodologies which assign main engine power according to ship speed, size and other characteristics. Fuel type would most likely be 0.1% MGO within the SECA unless scrubbers are used. The use of scrubbers by particular vessels can also be identified from certain databases. The paper is well written and the conclusions appear to be mostly justified although the contention that the plume identification worked well and that better meteorological data were not needed could questioned because of the reasons given above. The question also remains as to the usefulness of the measurements if only particle number concentration showed distinct peaks. More discussion as to the reasons for only particle number concentration showing distinct peaks would be useful. This reviewer does not have sufficient expertise to evaluate the particle measurement techniques.

---

## Referee Comment (RC2) · Anonymous Referee #2 · 11 Apr 2019

The paper provides a practical and simplistic approach to attribute particle emissions from ships, measured in the plume of the ships identified by AIS. The methodology is practical but there may be some uncertainties in the method. I provide my detailed comments below. I recommend a major revision.

You can refer to two more studies in the literature that specifically focused on ship plumes and characterization of emissions in northern latitudes. These are given below and may be added to the literature review section:

Aliabadi, A. A., Staebler, R. M. & Sharma, S. (2015), 'Air Quality Monitoring in Communities of the Canadian Arctic During the High Shipping Season with a Focus on

[Figure]

Local and Marine Pollution', Atmospheric Chemistry and Physics, 15(5), 2651-2673, doi: 10.5194/acp-15-2651-2015.

Aliabadi, A. A., Thomas, J. L., Herber, A. B., Staebler, R. M., Leaitch, R. W., Schulz, H., Law, K. S., Marelle, L., Burkart, J., Willis, M. D., Bozem, H., Hoor, P. M., Kollner, F., Schneider, J., Levasseur, M., & Abbatt, J. P. D. (2016), 'Ship Emissions Measurement in the Arctic from Plume Intercepts of the Canadian Coast Guard Icebreaker Amundsen from the Polar 6 Aircraft Platform', Atmospheric Chemistry and Physics, 16(12), 7899-7916, doi: 10.5194/acp-16-7899-2016.

The most major concern is lack of accurate wind measurements. It is likely the air circulation patterns near coastal areas be very non-uniform horizontally. For instance wind speeds and directions can change significantly from the location of the ship to that of the weather station on land. I understand that the simplistic nature of the method justifies using a few weather stations, but the authors can investigate potential errors more. Below are some ideas.

Was there wind speed and direction measurement on board of some ships? In this way you can characterize some differences between wind conditions on the sea and on land. You can also perform some hypothetical plume dispersion simulations near the coastal waters of interest to see if wind conditions are generally horizontally homogenous. You can use HYSPLIT web-based trajectory or dispersion modelling to investigate this quickly. For instance try some diurnal times and different seasons to investigate this. If you use trajectory modelling, you can investigate trajectories of air parcels arriving at the weather station of AQ trailer. Otherwise, if you use dispersion modelling, you can use point source and the ship stack to see where the plume goes. Having a few simple simulations included in the paper can add value on adequacy of the simplistic approach for meteorological model. (https://www.ready.noaa.gov/HYSPLIT.php)

The paper is already very short. So why not including all the supplemental figures, tables, text, and references in the main paper? This way the paper will be much easier

to read without having to refer to multiple documents.

The authors can compare their aerosol size distribution as a function of plume age to those reported by Aliabadi et al. 2015.

I hope these suggestions can improve the quality and demonstrate the suitability of the simplistic approach taken.

———————————————

---

## Author Comment (AC1) · 9 May 2019

We are thankful for the comments by referee no. 1, which has resulted in a more detailed description of both limitations and opportunities of our new method in the discussion section.

Referee comment 1.

The reference list is rather short and the Introduction too brief. It should cover expected characteristics of particles and other components of ship plumes (including particle size distribution and plume dispersion) as well as SECA regulations including fuel S content

and possible use of scrubbers.

Author's response 1.

A reason for having a rather short introduction was, in addition to trying to have a compact paper in general, that there is a lot of variation in the literature, both when it comes to results, methods, and scenarios. For example, a study of an aerosol property in a limited location during a limited period can differ from the results in another study, and it can be difficult to compare. This is also a motivation for doing our current study where we try to include many different aerosol measurements simultaneously and where the method has the potential to quantify the environmental impact of an arbitrary number of individual ships. We can however see how some more elaborate examples from the literature can be useful for the reader, especially if one is not very familiar with ship emission studies. Also, we think it can be good to highlight the variations in the literature even if it is not possible to directly compare those studies with ours. There was a similar comment from Referee 2, and we have added a section in the manuscript which addresses the concerns of both referees.

Author's changes in manuscript 1.

These are the following additions to the manuscript in the section "1 Introduction" (see corresponding reference list at the end of this response): "In the International Convention for the Prevention of Marine Pollution from Ships (MARPOL) Annex VI, the main exhaust gas emissions of sulphur oxides (SOx) and nitrous oxides (NOx) are limited. Hence, the international Maritime Organisation (IMO) have regulated the fuel sulphur content in several steps, with a total decrease from 1.5% to 0.1% mass fraction between the years 2010 and 2015 in Sulphur Emission Control Areas (SECA). In 2016 it was decided that further reduction of the fuel sulphur limit is going to be implemented, with a cap of 0.50 % sulphur in fuel oil on board all ships from January 1st 2020. A recent report showed a compliance level to the sulphur regulations of 92-94 % during 2015 and 2016 in the region around Denmark (within the Baltic Sea SECA). (Mellqvist

et al., 2017) Hence it is expected that most ships in the region are using fuels with a sulphur content of maximum 0.1 %. In addition to cleaner fuels, such as low-sulphur residual marine fuel oil, marine diesel oil (MDO), or liquefied natural gas (LNG), ships can comply by being equipped with scrubbers which remove the SO2 from the flue gas. The use of scrubbers was also observed in the region during our period of interest, by Mellqvist et al. (2017)."

"Particle number size distributions have been studied in atmospheric conditions previously, showing some variations in sizes and number of modes. This can be expected since many factors affect the emissions, such as engine operations, and the atmospheric transformation processes. For example, Jonsson et al. (2011) showed that size resolved particle number emission factors were largest around particle diameters of 35 nm, with smaller sizes observed for ships running on gas turbines than on diesel engines. Out of these particles, 36-46 % were non-volatile, and could contain some black carbon (BC). These measurements are from 2010, i.e. during the 1 % Sulphur limit within SECAs. Pirjola et al. (2014) showed that the number size distribution had two modes for fresh ship plumes, a dominating mode peaked at 20- 30 nm, and an accumulation mode at 80-100 nm. About 30 % of these were non-volatile, and it was also shown that the after treatment system affected the total particle number emission. These measurements are from 2010-2011. Diesch et al. (2013) observed a nucleation mode in the 10-20 nm diameter range and a combustion aerosol mode centred at about 35 nm. No particles with sizes above 1  $\mu$ m were observed. Six percent of the particle mass was due to BC. Other measurements on-board on a ship showed particle size distributions major peak at around 10 nm and a smaller peak at around 30-40 nm. Ca 40 % of the mass was non-volatile material, but particles below 10 nm consisted of only volatile material. (Hallquist et al., 2013) Westerlund et al. (2015) measured ship plumes from a stationary site and used AIS to characterise ships. Westerlund et al. found unimodal particle number size distributions for cargo and passenger ships, with the peak around 40 nm, while e.g. tug-boats emitted smaller particles. Since the measurements were carried out in a harbour area, as most of the other studies above, they

could capture changes in emissions during e.g. acceleration of ships. These harbour measurements were carried out in 2010, i.e. also before the 2015 SECA implementation. In another harbour area, Donateo et al. (2014) guantified the contribution of ship emissions to local total aerosol concentrations. The ship contribution to particle number was found to be 26 %. They could also see plume peaks in PM2.5, since measurements were done in a harbour area and plume peak concentrations were relatively high. A study performed in an Arctic region, showed a size distribution mode with peak around 27 nm during the first 6 hours of plume transport and later (>6 h) modes above 100 nm become more prominent. (Aliabadi et al., 2015) Here, the ship contribution to BC was estimated to be 4.3-9.8 %. Due to the clean Arctic environment and low background concentrations, the evolution of a ship plume contribution could be studied over time (0-72 h). Dispersion modelling of ship plumes has shown that dilution and coagulation are important processes within the first hour after emission, reducing the number concentration by four orders of magnitude and one order of magnitude, respectively. (Tian et al., 2014) The decrease in particle number concentration is most rapid during the first minutes after emission."

We have slightly modified the second last paragraph in the introduction section as follows to better accommodate with the new text above: "We present a new revised method to identify individual aerosol ship plumes based on AIS data and non-linear wind transport of the ship plume to a stationary coastal field site, which is several km downwind. The method has been tested on particle number concentration, particle number size distribution and black carbon mass. Also CO2, NOx, and aerosol mass spectrometry data is presented in the companion paper by Ausmeel et al. (Ship plumes in the Baltic Sea Sulphur Emission Control Area: Chemical characterization and contribution to coastal aerosol concentrations, manuscript in preparation, 2019b). The measurements were performed in Falsterbo, in southern Sweden, located downwind of a heavily trafficked shipping lane in the Oresund Strait with a daily average of 73 and 63 AIS transmitting ships passing in winter and summer respectively, and which connects the Atlantic and the Baltic Sea. The distance from the shipping lane to the

site corresponds to an average transport time of between 15 and 70 minutes (10-90th percentile) for the ship plumes. The measurements took place during the winter (Jan-Feb) and the summer (May-Jul) of 2016. With the new revised plume identification method, we can detect several tens of plumes in a day with favourable wind conditions. We also show how particle number concentration data can be used when AIS data is failing or missing, to identify individual ship plumes, however without information about which ship it is."

**Referee comment 2.**

P 8 last para – why is there no correlation with ship size? While the interpolation of hourly wind data to one minute yielded quite good correlation between ships identified by AIS and arrival of detected plumes at the measurement station, more detailed meteorological data might have allowed estimation of the extent of dispersion of the ship plumes and the location of the core of the plumes relative to the measurement station. This may have allowed better correlation between ship size and detected plume. For instance, the main part of a plume from a large ship might pass some distance from the measurement station. This is likely the reason for the poor correlation between ship size and size of particle number peaks.

**Author's response 2.**

Here, there is a misunderstanding. Our intended message was not clear and we appreciate that the referee raised this point. We have revised the text, clarifying that almost every ship that influences the measurement station contributes to the station with its entire ship plume in the horizontal extent. In other words, also with the core of the plume. For example, 10 km to the west of the station, a vast majority of the ships sail in a north-south or south-north direction in the current shipping lane. If the westerly winds reaching the measurement station 10 km to the east of the shipping lane are fairly stable during the time period of the passage of the ship in the lane, the

ship has go give away all parts of its shipping plume to the measurement station. This includes both the core of the plume and the lower concentrations of the plume. It is only in exceptional situations that the wind is changing drastically during a ship passage, meaning the core of the plume could potentially "miss" the station. However, we do not claim to know anything about the concentrations as function of altitude, and this anyhow doesn't matter a lot since we are mainly interested in the exposure to people at breathing height, and not at higher altitudes. Nevertheless, higher concentrations in higher vertical layers can potentially reach ground level further inlands during specific meteorological situations, which we do not account for.

**Author's changes in manuscript 2.**

We have clarified in the manuscript in the section "3.1 Ship plume identification and analysis" that almost all ships contribute with their core of the plume at the station: "In theory, it is possible that the wind direction is changing as the ships sail past the measurement station, meaning that we can potentially miss the maximum concentration in ship plumes, and only record the lower concentrations at the tails of the ship plumes. However, in almost all cases in our data set, the wind is stable enough during each ship plume passage at the station. This means, we fetch entire ship plumes, from the lowest concentrations in the plumes to the maximum concentrations in the plume".

**Referee comment 3.**

At Line 30 it is stated that there was no information on engine operation and fuel. However, this can be derived from the AIS data according to well established methodologies which assign main engine power according to ship speed, size and other characteristics. Fuel type would most likely be 0.1% MGO within the SECA unless scrubbers are used. The use of scrubbers by particular vessels can also be identified from certain databases.

Author's response 3.

It is good that the referee points out that this kind of data is available. Our sentence in the manuscript is unclear and can be interpreted as if there is no such data available in general. We agree with the referee that the fuel mentioned (0.1 % S) should be the most common, with the potential exception if someone is not following the regulation. A recent report showed an average compliance of 94 % in the region during 2015 and 2016. (Mellqvist et al., 2017) Based on information on the total power of the ships, their design speed, and actual speed when passing the measurement site (information retrieved from AIS), we have used the propeller law to retrieve an estimate of the engine power required. If the contribution to particle number concentrations from ships are dependent on engine power required, we should see a relationship when we plot particle number concentration contribution as function of engine power for different ships. Since we cannot control for e.g. meteorological conditions and the vertical plume dispersion we chose to pick out plumes from a winter day with constant wind direction and ships at similar distance from the site, and compare these to the engine power of the ships (i.e. "power in use", based on design speed and actual speed). The result is shown in Figure 1, for ships sailing in the shipping lane west of the measurement station during January 28, 2016. There were 13 plumes that could be clearly assigned to an individual ship (other occasions where several plumes overlapped also existed but were not considered) for which there was also information on engine power and design speed. From this graph, there is not much more dependence of PN on engine power than we could previously see when just comparing with e.g. ship size. Hence, we chose not to go further with this kind of analysis for this manuscript. There could be an effect visible if a larger ship sample size can be studied (longer measurements required), but there is also reason to believe that a large dependence between PN emissions and engine power is not to be expected at our measurement site. Firstly, the ships in our region of study are limited in size, since the largest ships are not sailing in the Oresund Strait. That is, we do not expect several orders of magnitude of difference in engine power between the ships in our sample (which was also confirmed by the AIS data). Secondly, even though particle mass (PM) concentration emissions are

generally higher with higher engine load, this is not necessarily true for particle number concentration emissions. That is, a plume with high number concentration of particles can have a lower mass concentration than a plume with fewer particles. PN emission factors can be found in the literature, but there is no clear agreement how the engine loads and different fuels impact PN emissions. However, this is not to say that such a trend can be found in other conditions, for example if studying a larger span of the international ship fleet and a comparison between ship type. And for studies focusing on PM (which our presented method can be used for) such dependencies can also be studied. Since our aim with this manuscript is mainly to describe the methods of plume identification and analysis (identifying ship plumes using particle measurements, connecting these plumes to individual ships using AIS, and to use the knowledge of a plume passage to analyses other aerosol data such as black carbon concentration) we are not mainly interested in looking deeper into the technical characteristics of the ships and the connection to emissions. However, this is still an important question and should be the topic of other studies and publications in the future. To make it more clear to the reader what is possible to do, we have added an additional section about this in the "Recommendations and concluding remarks" section.

Author's changes in manuscript 3.

We have added a section in the section "5 Recommendations and concluding remarks": "In future studies of detailed individual ship plumes and the emission sources, it should be considered whether the particle emissions depend on ship engine power used. It is possible to estimate the engine power required by a ship, using the total power of the ships, their design speed, and actual speed through the propeller law. (Moreno-Gutiérrez et al., 2015) This can then be compared to particle number concentration emissions, but also particle mass emissions and gaseous emissions. With the method presented in this paper, it is possible to collect information on a very large sample of ships for these kinds of investigations." We have also removed from section "4.1 Plume identification": "The reason for the lack of correlation between emissions and e.g. ship size could be the heterogeneity in ship and meteorological parameters. Emissions also depend on e.g. engine operation and fuel, which we do not have information on. We show that a sample of a hundred plumes was not enough to find such relations, if they existed, at this distance from the ships."

Referee comment 4.

The question also remains as to the usefulness of the measurements if only particle number concentration showed distinct peaks. More discussion as to the reasons for only particle number concentration showing distinct peaks would be useful.

**Author's response 4.**

The referee is correct that plumes from various measurement parameters are not always visible in the measurements, although there is a particle number concentration peak clearly visible at the same time. This is however not an indication that the method does not work, but merely proves that the ship emissions do not influence this measurement parameter to a large extent, which is an important method outcome in itself. But, if particle number concentration peaks are not visible, we know that we are too far away from the ships to be able to detect individual plume peaks, and we see only an enhanced background concentration, which we cannot easily transform to a contribution from ships. Hence, a particle counter is very much needed to test whether the current method works, but the peaks of other parameters do not have to appear. Since this have created some misunderstanding, we are grateful to the referee to raise this point, and we have tried to clarify this in the manuscript. Additionally, even if the plumes are not distinguishable for the naked eye, e.g. in the BC time series, the knowledge of the time when the plume reached and passed the site could be used to get an estimate of the BC contribution. Of course, the uncertainty will be larger due to the low levels compared to the background and the signal-to-noise ratio, but we could show that there was indeed a significant difference between the background and the ship plume event for BC. Also, our study is performed at a single site and for studies in other conditions

and distances to the shipping lane, there can be higher or lower plume concentrations than in this study. As we mentioned above, low contribution results are also valuable as with the BC contribution. The observed ships are apparently not contributing very much in relative terms to the local BC levels. And there can potentially be other parameters where the ships contribute even less to the particle concentrations, even with no significant contribution. In this paper we focus on the methodology, PN concentrations and BC. We also have prepared a manuscript with a different angle in which among other things sulphate content and NOx measurements are presented. We will refer to this paper for further reading about the contribution of other measurement parameters.

Author's changes in manuscript 4.

We have previously mentioned that BC concentrations cannot be seen as plumes, and other measurement parameters might not be viewed as clear plume peaks either, but it is still possible to estimate a contribution to particle concentrations with one of the proposed new methods. To clarify this further, we have added a sentence in "5 Recommendations and concluding remarks" which deals with the recommendations to future users of the methods developed: Old text: "The method to estimate plume contribution from individual ships proved to be straightforward for the clearly visible ship plumes at the measurement station. For the eBC concentration, the plume identification was less straightforward since the plume signal was very low relative to the noise level. For many plumes, no increase in eBC was observed with the bare eye. We still used the already identified plumes to calculate the contribution to eBC." New addition to the text: "The method to estimate plume contribution from individual ships proved to be straightforward for the clearly visible ship plumes at the measurement station. For the eBC concentration, the plume identification was less straightforward since the plume signal was very low relative to the noise level. For many plumes, no increase in eBC was observed with the bare eye. We still used the already identified plumes to calculate the contribution to eBC. A very low, but still significant plume contribution could be calculated. Even if the proposed method yields non-significant plume contributions

for a specific parameter, this does not mean that the method does not work. Rather it means that ship emissions do not contribute to significant exposure inland for this parameter, and that the detection capabilities of the instrument do not allow for detecting this non-significant contribution."

**References:**

Aliabadi, A. A., Staebler, R. M., and Sharma, S.: Air quality monitoring in communities of the Canadian Arctic during the high shipping season with a focus on local and marine pollution, Atmos. Chem. Phys., 15, 2651-2673, 10.5194/acp-15-2651-2015, 2015. Diesch, J.-M., Drewnick, F., Klimach, T., and Borrmann, S.: Investigation of gaseous and particulate emissions from various marine vessel types measured on the banks of the Elbe in Northern Germany, Atmos, Chem. Phys., 13, 3603-3618. 2013. Donateo, A., Gregoris, E., Gambaro, A., Merico, E., Giua, R., Nocioni, A., and Contini, D.: Contribution of harbour activities and ship traffic to PM2.5, particle number concentrations and PAHs in a port city of the Mediterranean Sea (Italy), Environmental Science and Pollution Research, 21, 9415-9429, 10.1007/s11356-014-2849-0, 2014. Hallquist, Å. M., Fridell, E., Westerlund, J., and Hallquist, M.: Onboard Measurements of Nanoparticles from a SCR-Equipped Marine Diesel Engine, Environ. Sci. Technol., 47, 773-780, 10.1021/es302712a, 2013. Jonsson, Å. M., Westerlund, J., and Hallquist, M.: Size-resolved particle emission factors for individual ships, Geophys. Res. Lett., 38, doi:10.1029/2011GL047672, 2011. Mellqvist, J., Beecken, J., Conde, V., and Ekholm, J.: Surveillance of Sulfur Emissions from Ships in Danish Waters, 2017. Moreno-Gutiérrez, J., Calderay, F., Saborido, N., Boile, M., Rodríguez Valero, R., and Durán-Grados, V.: Methodologies for estimating shipping emissions and energy consumption: A comparative analysis of current methods, Energy, 86, 603-616, https://doi.org/10.1016/j.energy.2015.04.083, 2015. Pirjola, L., Pajunoja, A., Walden, J., Jalkanen, J.-P., Rönkkö, T., Kousa, A., and Koskentalo, T.: Mobile measurements of ship emissions in two harbour areas in Finland, Atmospheric Measurement Techniques, 7, 149-161, 2014. Tian, J., Riemer, N., West, M., Pfaffenberger, L., Schlager, H., and Petzold, A.: Modeling the evolution of aerosol particles in a ship plume using PartMC-MOSAIC, Atmos. Chem. Phys., 14, 5327-5347, 10.5194/acp-14-5327-2014, 2014. Westerlund, J., Hallquist, M., and Hallquist, Å. M.: Characterization of fleet emissions from ships through multi-individual determination of size-resolved particle emissions in a coastal area, Atmos. Environ., 112, 159-166, https://doi.org/10.1016/j.atmosenv.2015.04.018, 2015.

Fig. 1. Total particle number concentration within ship plumes plotted versus calculated ship engine power in use, from measurements during January 28 2016.

---

## Author Comment (AC2) · 9 May 2019

We appreciate the comments by the referee, especially about wind uncertainties for our methodology, e.g. the impact of sea breeze, which was not explored in the manuscript. Now we have addressed this issue as described below.

Referee comment 1.

You can refer to two more studies in the literature that specifically focused on ship plumes and characterization of emissions in northern latitudes. These are given below and may be added to the literature review section: Aliabadi, A. A., Staebler, R. M.

& Sharma, S. (2015), 'Air Quality Monitoring in Communities of the Canadian Arctic During the High Shipping Season with a Focus on Atmospheric Chemistry and Physics, 15(5), 2651-2673, doi: 10.5194/acp-15-2651-2015. Aliabadi, A. A., Thomas, J. L., Herber, A. B., Staebler, R. M., Leaitch, R. W., Schulz, H., Law, K. S., Marelle, L., Burkart, J., Willis, M. D., Bozem, H., Hoor, P. M., Kollner, F., Schneider, J., Levasseur, M., & Abbatt, J. P. D. (2016), 'Ship Emissions Measurement in the Arctic from Plume Intercepts of the Canadian Coast Guard Icebreaker Amundsen from the Polar 6 Aircraft Platform', Atmospheric Chemistry and Physics, 16(12), 7899- 7916, doi: 10.5194/acp-16-7899-2016.

Author's response 1.

We acknowledge that we have not covered the complete literature when it comes to ship emission studies, since this is a large and wide subject (see also response to Referee 1, comment 1). The suggested papers are interesting. The first can be a good comparison between our results and a clean environment as the Arctic. The second paper presents airborne plume measurements, which we have not discussed in detail, but as is also mentioned in Referee Comment 4, the ship plume particle size distribution as a function of age is possibly comparable to some of the plumes in Falsterbo. All plumes in our manuscript are very fresh compared to the results in Aliabadi et al. 2016. We consider that the first paper suggested by the referee (Aliabadi et al., 2015) is the most relevant to our study and have chosen to include it in the section "1. Introduction section" together with several other papers relevant to our own work. There was a similar comment from Referee 1, and we have added a section in the manuscript which addresses the concerns of both referees.

Author's changes in manuscript 1.

"Particle number size distributions have been studied in atmospheric conditions previously, showing some variations in sizes and number of modes. This can be expected since many factors affect the emissions, such as engine operations, and the atmo-

spheric transformation processes. For example, Jonsson et al. (2011) showed that size resolved particle number emission factors were largest around particle diameters of 35 nm, with smaller sizes observed for ships running on gas turbines than on diesel engines. Out of these particles, 36-46 % were non-volatile, and could contain some black carbon (BC). These measurements are from 2010, i.e. during the 1 % Sulphur limit within SECAs. Pirjola et al. (2014) showed that the number size distribution had two modes for fresh ship plumes, a dominating mode peaked at 20– 30 nm, and an accumulation mode at 80–100 nm. About 30 % of these were non-volatile, and it was also shown that the after treatment system affected the total particle number emission. These measurements are from 2010-2011. Diesch et al. (2013) observed a nucleation mode in the 10–20 nm diameter range and a combustion aerosol mode centred at about 35 nm. No particles with sizes above 1 $\mu$m were observed. Six percent of the particle mass was due to BC. Other measurements on-board on a ship showed particle size distributions major peak at around 10 nm and a smaller peak at around 30−40 nm. Ca 40 % of the mass was non-volatile material, but particles below 10 nm consisted of only volatile material. (Hallquist et al., 2013) Westerlund et al. (2015) measured ship plumes from a stationary site and used AIS to characterise ships. Westerlund et al. found unimodal particle number size distributions for cargo and passenger ships, with the peak around 40 nm, while e.g. tug-boats emitted smaller particles. Since the measurements were carried out in a harbour area, as most of the other studies above, they could capture changes in emissions during e.g. acceleration of ships. These harbour measurements were carried out in 2010, i.e. also before the 2015 SECA implementation. In another harbour area, Donateo et al. (2014) quantified the contribution of ship emissions to local total aerosol concentrations. The ship contribution to particle number was found to be 26 %. They could also see plume peaks in PM2.5, since measurements were done in a harbour area and plume peak concentrations were relatively high. A study performed in an Arctic region, showed a size distribution mode with peak around 27 nm during the first 6 hours of plume transport and later (>6 h) modes above 100 nm become more prominent. (Aliabadi et al., 2015) Here, the ship contribution to

BC was estimated to be 4.3-9.8 %. Due to the clean Arctic environment and low background concentrations, the evolution of a ship plume contribution could be studied over time (0-72 h). Dispersion modelling of ship plumes has shown that dilution and coagulation are important processes within the first hour after emission, reducing the number concentration by four orders of magnitude and one order of magnitude, respectively. (Tian et al., 2014) The decrease in particle number concentration is most rapid during the first minutes after emission."

Referee comment 2.

The most major concern is lack of accurate wind measurements. It is likely the air circulation patterns near coastal areas be very non-uniform horizontally. For instance wind speeds and directions can change significantly from the location of the ship to that of the weather station on land. I understand that the simplistic nature of the method justifies using a few weather stations, but the authors can investigate potential errors more. Below are some ideas. Was there wind speed and direction measurement on board of some ships? In this way you can characterize some differences between wind conditions on the sea and on land. You can also perform some hypothetical plume dispersion simulations near the coastal waters of interest to see if wind conditions are generally horizontally homogenous. You can use HYSPLIT web-based trajectory or dispersion modelling to investigate this quickly. For instance try some diurnal times and different seasons to investigate this. If you use trajectory modelling, you can investigate trajectories of air parcels arriving at the weather station of AQ trailer. Otherwise, if you use dispersion modelling, you can use point source and the ship stack to see where the plume goes. Having a few simple simulations included in the paper can add value on adequacy of the simplistic approach for meteorological model. (https://www.ready.noaa.gov/HYSPLIT.php)

Author's response 2.

The referee has raised a very important point about meteorology and winds, which affects the applicability of the method, and we appreciate the two alternative suggestions to check how meteorology can affect our measurements. We have decided to follow the suggestions with air mass back trajectories. We argue that sea breeze situations are among the most extreme situations when wind measurements in an erroneous way can show that we have winds from a shipping lane, while in reality the air does not come from the shipping lane. During a sea breeze, the local wind direction can be reversed as compared to the large scale air flow. We decided to investigate sea breezes at our measurement site in Falsterbo, and also insert some recommendations in the manuscript for coastal measurements when sea breeze is a common phenomenon. During the development of a sea breeze, the winds close to the ground level at the shore line can slowly start to blow towards land due to land warming up more than the sea from solar light absorption, despite that the large scale circulation is showing a different wind direction. At the start-up phase, the winds closest to the shore line where we perform our wind measurements at our measurement station will then not agree with the winds over the shipping lane further away from the shore line. Hence, the ship plumes might not reach the measurement station despite that the measured wind direction is suggesting that. In this situation, our wind path method will not work, and we will not register an enhanced ship plume concentration at the station. The danger in this situation is that this is interpreted as if the ships are contributing negligible pollution to our measurement site, while in reality the winds from the ships have not even reached the station. On the other hand, with a fully developed sea breeze later in the afternoon, the horizontal extension of the winds going from the sea towards inland locations have increased and the ship plumes can potentially reach the measurement station again, and the method works once more. Fully developed sea breezes can have horizontal extensions on the order of more than 50 kilometers (Pokhrel and Lee, Atmospheric Pollution Research, 2011, 2, 106-115). Land breezes can also potentially be a problem for the analysis, although land breezes are normally weaker than sea breezes. When we checked if there were sea breeze situations in Falsterbo, we have discovered that indeed there are two different occasions (in total 4 days) when there

were sea breezes developing during lunch time until late afternoon in May 2016. The local wind measured at the station indicated that we should be receiving air from the shipping lane to the west, while synoptic surface pressure wheather maps and our Hysplit trajectory analysis showed wind directions coming from the north. Unfortunately, we could not investigate how this affected the ship plume analysis at Falsterbo, since our instrumentation did not work during these sea breeze periods. There were no other strong sea breeze periods during the remaining period of our summer measurements. The sea breeze situations in Falsterbo could have disqualified our wind method, at least before the sea breeze became fully developed later during the day, and when the sea breeze started to fade out. With a continental area with a larger contrast in temperatures between land and sea and with larger continental and sea area, these problems can be even more common than in Falsterbo. There can also be other meteorological situations when the wind appears to be coming from the shipping lane, while in reality it is not. In summary, before performing ship plume measurements, each measurement location should be investigated for the occurrence of sea breezes and their horizontal extension, to be able to set up the experiments in a suitable way. However, the sea breeze problem and other meteorological phenomenon should not disqualify any measurement location. Namely, sea breezes do not take place all the time, and even during sea breezes, we might record shipping plumes at the shore line as we explained above. Nevertheless, users of the current method should be cautious to the occurrence of sea breezes. With this difficulty in mind, we have found that it is even more important to bring a particle counter to the measurement site. Namely, if the wind measurements are showing that the ships should be affecting the air quality at the measurement site, but the particle counter is not measuring any detectable plume for any ships, this indicates that the winds from the ships are not reaching the measurement site. Author's changes in manuscript 2. We have added a recommendation in the section "5 Recommendation and concluding remarks": "Before performing the measurements with the new method, it is important to investigate the meteorological situations at the current measurement site. For example, during sea breezes, local

wind measurements could indicate that shipping lane emissions should reach the measurement station, whereas in reality they might not. Care should be taken to account for these periods when the meteorological data will give erroneous results. However, these meteorological phenomena do not take place all the time, hence these specific meteorological conditions will not disqualify any chosen measurement site with the current proposed method. Again, these uncertain wind conditions make it very important to bring a particle counter to register shipping plumes. If the particle counter does not register any ship plumes during a selected time period, this indicates that winds from the ships are not reaching the measurement station, despite that the local wind measurements are suggesting otherwise."

Referee comment 3.

The paper is already very short. So why not including all the supplemental figures, tables, text, and references in the main paper? This way the paper will be much easier to read without having to refer to multiple documents.

Author's response 3.

We do not have a strong opinion against this suggestion. The placement of some information in the supplement was mainly an attempt to make the paper short and concise. But we agree with the point provided by the referee that it will not be practical for readers to look up information in the supplement. And if readers are interested, it can be a good idea to have all information easily accessible, since the manuscript is rather short at the moment. Re-considering the division between the manuscript and the supplement, we think that all information suits well in the original manuscript, except for the measurements of diffusion losses (Table S2 and Figure S2) due to its technical character. However, this is also such a short note that if the rest of the supplement is merged into the manuscript, it can be moved too, whereby we remove the supplement document entirely.

Author's changes in manuscript 3.

We have merged the supplementary information into the manuscript. The sections on measurement setup, meteorological parameters, and the characterisation of losses in the dryers is moved to "2. Instrumentation set-up and experimental site" and the section on log-normal parameters is moved to the section "4.2. Results of plume contribution calculations".

Referee comment 4.

The authors can compare their aerosol size distribution as a function of plume age to those reported by Aliabadi et al. 2015.

Author's response 4.

We have included the suggested scientific article in the section "1 Introduction", together with references to several other studies of size resolved particle number emissions from ships. Therefore, we have not included a specific comparison with the suggested size distribution, but rather given a broader background to the field to the reader. For comparing size distributions in detail, one has to consider the various differences between the many existing studies, which is relevant and possible to do, but outside of the main message of our manuscript. We are mainly interested in reporting the contribution to Falsterbo. But for future investigations the size distribution during longer transport times, it will be of value to compare with the suggested paper. Author's changes in manuscript 4. See "Author's changes in manuscript 2."

References:

Aliabadi, A. A., Staebler, R. M., and Sharma, S.: Air quality monitoring in communities of the Canadian Arctic during the high shipping season with a focus on local and marine pollution, Atmos. Chem. Phys., 15, 2651-2673, 10.5194/acp-15-2651-2015, 2015. Diesch, J.-M., Drewnick, F., Klimach, T., and Borrmann, S.: Investigation of gaseous and particulate emissions from various marine vessel types measured on the banks of the Elbe in Northern Germany, Atmos. Chem. Phys., 13, 3603-3618,

2013. Donateo, A., Gregoris, E., Gambaro, A., Merico, E., Giua, R., Nocioni, A., and Contini, D.: Contribution of harbour activities and ship traffic to PM2.5, particle number concentrations and PAHs in a port city of the Mediterranean Sea (Italy), Environmental Science and Pollution Research, 21, 9415-9429, 10.1007/s11356-014-2849-0, 2014. Hallquist, Å. M., Fridell, E., Westerlund, J., and Hallquist, M.: On-board Measurements of Nanoparticles from a SCR-Equipped Marine Diesel Engine, Environ. Sci. Technol., 47, 773-780, 10.1021/es302712a, 2013. Jonsson, Å. M., Westerlund, J., and Hallquist, M.: Size-resolved particle emission factors for individual ships, Geophys. Res. Lett., 38, doi:10.1029/2011GL047672, 2011. Mellqvist, J., Beecken, J., Conde, V., and Ekholm, J.: Surveillance of Sulfur Emissions from Ships in Danish Waters, 2017. Moreno-Gutiérrez, J., Calderay, F., Saborido, N., Boile, M., Rodríguez Valero, R., and Durán-Grados, V.: Methodologies for estimating shipping emissions and energy consumption: A comparative analysis of current methods, Energy, 86, 603-616, https://doi.org/10.1016/j.energy.2015.04.083, 2015. Pirjola, L., Pajunoja, A., Walden, J., Jalkanen, J.-P., Rönkkö, T., Kousa, A., and Koskentalo, T.: Mobile measurements of ship emissions in two harbour areas in Finland, Atmospheric Measurement Techniques, 7, 149-161, 2014. Tian, J., Riemer, N., West, M., Pfaffenberger, L., Schlager, H., and Petzold, A.: Modeling the evolution of aerosol particles in a ship plume using PartMC-MOSAIC, Atmos. Chem. Phys., 14, 5327-5347, 10.5194/acp-14-5327-2014, 2014. Westerlund, J., Hallquist, M., and Hallquist, Å. M.: Characterization of fleet emissions from ships through multi-individual determination of size-resolved particle emissions in a coastal area, Atmos. Environ., 112, 159-166, https://doi.org/10.1016/j.atmosenv.2015.04.018, 2015.